# A Protein Misfolding Shaking Amplification-based method for the spontaneous generation of hundreds of bona fide prions

Hasier Eraña [1,2,3], Cristina Sampedro-Torres-Quevedo [1], Jorge M. Charco [1,2,3], Carlos M. Díaz-Domínguez [1,2], Francesca Peccati[1], Maitena San-Juan-Ansoleaga [1], Enric Vidal [4,5], Nuno Gonçalves-Anjo [1], Miguel A. Pérez-Castro [1], Ezequiel González-Miranda[1], Patricia Piñeiro [1], Leire Fernández-Veiga [1], Josu Galarza-Ahumada [1], Eva Fernández-Muñoz[1], Guiomar Perez de Nanclares [6], Glenn Telling[7], Mariví Geijo [8], Gonzalo Jiménez-Osés [1,9] & Joaquín Castilla [1,2,9] ✉

Prion diseases are a group of rapidly progressing neurodegenerative disorders caused by the misfolding of the endogenous prion protein (PrP$^C$) into a pathogenic form (PrP$^{Sc}$). This process, despite being the central event underlying these disorders, remains largely unknown at a molecular level, precluding the prediction of new potential outbreaks or interspecies transmission incidents. In this work, we present a method to generate bona fide recombinant prions de novo, allowing a comprehensive analysis of protein misfolding across a wide range of prion proteins from mammalian species. We study more than 380 different prion proteins from mammals and classify them according to their spontaneous misfolding propensity and their conformational variability. This study aims to address fundamental questions in the prion research field such as defining infectivity determinants, interspecies transmission barriers or the structural influence of specific amino acids and provide invaluable information for future diagnosis and therapy applications.

Transmissible spongiform encephalopathies (TSEs), or prion diseases, are inevitably lethal and rapidly progressing neurodegenerative disorders that so far have been observed in humans and other mammals. The underlying cause of these diseases is the misfolding of the cellular prion protein (PrP$^C$), present in all mammalian species, resulting in an aberrantly folded, aggregation-prone, neurotoxic form referred to as PrP$^{Sc}$ [1]. While the cellular protein is soluble, characterized by α-helix domains, and susceptible to proteases, the pathogenic isoform is insoluble, enriched in β-sheet domains, partially resistant to proteases, neurotoxic, and most importantly, capable of inducing its abnormal conformation onto the cellular counterpart, leading to the formation of aggregates. This ability to induce the abnormal conformation in PrP$^C$ is the driving force behind the disease's spread within the central nervous system of affected

[1]Center for Cooperative Research in Biosciences (CIC BioGUNE), Basque Research and Technology Alliance (BRTA), Derio, Spain. [2]Centro de Investigación Biomédica en Red de Enfermedades infecciosas (CIBERINFEC), Carlos III National Health Institute, Madrid, Spain. [3]ATLAS Molecular Pharma S. L, Derio, Spain. [4]IRTA. Programa de Sanitat Animal. Centre de Recerca en Sanitat Animal (CReSA). Campus de la Universitat Autònoma de Barcelona (UAB), Bellaterra, Catalonia, Spain. [5]Unitat mixta d'Investigació IRTA-UAB en Sanitat Animal. Centre de Recerca en Sanitat Animal (CReSA). Campus de la Universitat Autònoma de Barcelona (UAB), Bellaterra, Catalonia, Spain. [6]Molecular (Epi)Genetics Laboratory, Bioaraba Health Research Institute, Araba University Hospital, Vitoria-Gasteiz, Spain. [7]Prion Research Center, Colorado State University, Fort Collins, CO, USA. [8]Animal Health Department, NEIKER-Basque Institute for Agricultural Research and Development. Basque Research and Technology Alliance (BRTA), Derio, Spain. [9]IKERBASQUE, Basque Foundation for Science, Bilbao, Spain. ✉e-mail: jcastilla@cicbiogune.es

individuals, triggering neurodegeneration and serving as a pivotal factor in inter-individual transmission[2].

Depending on the origin of the misfolded protein, prion diseases can be classified as follows. Acquired forms result from exposure to exogenous PrP$^{Sc}$ through ingestion or medical procedures. Familial or genetic cases arise from mutations in the PrP$^{C}$-encoding gene (*PRNP*), which appear to increase its misfolding proneness into PrP$^{Sc}$. Idiopathic, commonly misnamed sporadic, occurrences involve rare, spontaneous misfolding of wild-type PrP$^{C}$. Despite the lack of knowledge on potential factors triggering the latter forms, idiopathic prion disorders are the most prevalent, accounting for 85–90% of human cases[3], and also observed in animals such as sheep and cattle[4,5]. Although prevalence studies in other mammals suffering sporadic prion diseases are highly challenging, it cannot be ruled out that they occur with a similar incidence to that in humans, estimated to be between 1 and 5 cases per million individuals per year[6].

During decades, the possibility of an exclusively protein-based infectious agent, as stated by the "protein only" hypothesis[1], has been a subject of controversy. However, the development of in vitro prion propagation methods able to reproduce the prion misfolding phenomena with minimum components in a controlled environment was pivotal to conclude the debate[7]. Nonetheless, the definitive demonstration of the theory took nearly three decades, as the complexity of initial propagation systems left room for alternative explanations for perplexing phenomena at the time, such as the existence of multiple prion strains[8] or highly selective interspecies transmission barriers[9]. Demonstrating such sophisticated biological properties for a single protein and an apparently straightforward misfolding event, required several sequential developments of in vitro prion propagation and generation methods.

Initial attempts to replicate the prion misfolding phenomenon in vitro, based on partially purified PrP$^{C}$, demonstrated that PrP$^{Sc}$ or prions could be formed from PrP$^{C}$ through a nucleation-dependent process, using PrP$^{Sc}$-seeded amplification reactions, thus mimicking acquired prion diseases[10,11]. While these reactions exhibited strain-specific propagation and the replication of interspecies transmission barriers[12,13], the conversion yields remained low, raising concerns within the community. Conversion efficiency was subsequently improved by using whole brain homogenates and incorporating cycles of incubation and sonication, leading to the development of Protein Misfolding Cyclic Amplification (PMCA)[14]. PMCA provided definitive evidence of nucleation-dependent prion propagation through serially diluted amplification reactions, generating infectious prions with properties similar to the original PrP$^{Sc}$ seed[15]. Additionally, PMCA helped uncover the role of unknown brain components, identified as conversion cofactors, as well as physical enhancers like sonication or the addition of beads to increase prion propagation efficiency[16,17], resulting in an extraordinarily efficient prion propagation system. Such efficiency on seed-induced prion misfolding eventually led to de novo prion misfolding in vitro[18,19], which mimics the main event in sporadic prion disorders. Despite this, prion formation remained infrequent and stochastic, likely due to factors such as variability in PrP$^{C}$ sources and challenging parameter control. In the absence of consistently reproducible systems that gave rise to spontaneously misfolded bona fide prions, the use of recombinant PrP (rec-PrP) was explored to simplify and enhance the study of this phenomenon. The first report of synthetic prion generation in vitro showed potential for disease induction in animal models[20], albeit requiring a truncated rec-PrP and a PrP overexpressing mouse model that also developed a neurological disease without inoculation. While several subsequent attempts brought improvements[18,21,22], a significant breakthrough occurred in 2010 with the spontaneous generation of highly infectious recombinant prions in wild-type mice using minimal components[23], providing definitive support for the "protein only" hypothesis, and offering a model for studying sporadic TSEs. However, the complexity of the required equipment and specific skills, together with the stochastic nature of spontaneous PrP misfolding hindered reproducibility, as evidenced by inconsistent results among research groups[24,25]. Consequently, despite the importance of spontaneous PrP misfolding as the central event in prion disease pathobiology, there was no method, which consistently reproduced this event in vitro with minimal components, allowing the systematic study of the phenomenon.

Our study reports a methodology grounded on the previously developed Protein Misfolding Shaking Amplification (PMSA) technique[26], which demonstrates the ability to consistently generate bona fide infectious prions de novo. In this work, we display the potential of this method by conducting a comprehensive analysis of spontaneous protein misfolding across a vast range of prion proteins obtained from over 700 mammalian species. After curation of redundant sequences, our research involved the testing of more than 380 different recombinant proteins, allowing us to unravel their misfolding propensity. Thus, apart from delving extensively into the PMSA technique, providing a comprehensive description of all necessary components required for achieving efficient and reproducible de novo protein misfolding in vitro, we also demonstrate the utility and reproducibility of the method by applying it to 382 distinct proteins. This tailored approach enables us to assess the misfolding propensity of each protein variant in a quantitative manner. An ordered scoring list that ranks the spontaneous misfolding propensity is accompanied by an in silico prediction of the thermostability of the globular form of selected variants to unveil potential correlation. Finally, we validate the authenticity of the prions generated using our methodology through the inoculation of selected misfolded proteins from various species in a highly susceptible animal model. In summary, we present a method that consistently allows the spontaneous misfolding of rec-PrP into bona fide prions using minimal components, validated through the generation of hundreds of prions for a wide variety of species, thus providing insight on the main event underlying the most prevalent prion diseases.

## Results
### Collection of mammalian prion protein sequences

To conduct the most comprehensive analysis of prion protein misfolding propensity determinants, we needed to gather the widest and most diverse possible collection of prion protein sequences. Given that so far TSEs have only been reported in certain mammalian species, our search for *PRNP* genes was focused on those of species from the class Mammalia. For that, three different sources were explored: (i) already annotated *PRNP* or prion protein sequences retrieved from GenBank, where we identified 301 sequences corresponding to distinct species. During this process, we excluded polymorphic variants and selected the most relevant sequence when available, focusing our efforts in this first stage on wild-type variants, although the inclusion of polymorphic variants is intended soon. (ii) Sequences extracted from whole genome sequencing projects [Sequence Read Archive (SRA) and DNAZoo], annotating 393 distinct *PRNP* sequences from raw sequencing data. (iii) DNA samples extracted from biological fluid and tissue samples of specimens kindly provided by various zoos and animal shelters. The open reading frame (ORF) of the *PRNP* gene for those species was sequenced using specifically designed primers based on closely related species with available PrP sequences. This latter approach allowed obtaining an additional 31 sequences, resulting in a total of 725 PrP sequences from as many mammal species. It is important to note that this dataset excludes polymorphic variants and is considered herein as a wild-type collection. As shown in Supplementary Table 1, PrP sequences from representatives of 27 out of 29 orders within the class Mammalia could be obtained (with representatives of 128 out of 153 families and 436 out of 1231 genera), achieving a robust sample of the variability of PrP sequences across mammals. This collection of sequences, for which a phylogenetic

analysis will be published elsewhere soon, is now publicly available in GenBank [National Center for Biotechnology Information (NCBI)].

Since one of our main interests lies on identifying motifs or residues within the PrP sequence with greater impact on spontaneous misfolding proneness and given that the flexible N-terminus of the protein is not part of the protease-resistant amyloid core of prions, we decided to compare sequences from residue 90 to 231 (based on bank vole PrP amino acid numbering, which is our gold standard). Focusing on this structured region of around 141 amino acids, depending on the species, all collected sequences were aligned and redundancies were eliminated. This refinement process resulted in a dataset of 382 PrP sequences, each unique in their non-disordered region (Supplementary Table 1).

### Generation of plasmids, recombinant protein expression, purification, and substrate preparation for in vitro misfolding

Once selected, plasmids were generated for the recombinant production of the distinct prion proteins in *Escherichia coli*. The ORF of each *PRNP* encoding PrP (23-C-ter end), which was sourced either from genomic DNA in biological tissues or fluids or obtained by synthesis, were cloned into the pOPIN E vector for their recombinant expression. For a few protein variants, expression yields were significantly lower than for the rest for undetermined reasons, this was solved by scaling up the culture volume. Upon elution, all purified proteins were concentrated at 25 mg/ml for the preparation of substrates for Protein Misfolding Shaking Amplification (PMSA). However, as recombinant PrP (rec-PrP) were preserved in a 6 M guanidine solution for long-term storage, they required dialysis to facilitate re-folding prior to substrate preparation. Following dialysis, each protein was diluted in conversion buffer, and dextran sulfate was added at a final concentration of 0.5% (w/v). Before assessing the misfolding propensity of each substrate, we ensured similar protein concentrations, through electrophoresis and total protein staining. Supplementary Fig. 1 shows 27 different substrates from a total of 382 prepared, with representative species for each distinct order. It highlights the similar concentrations, the different sizes of rec-PrP [mainly due to variations in the number of octapeptide repeats (OR), ranging from one OR (178 amino acids) to seven OR (233 amino acids). Amino acid number refers to the residues of each rec-PrP, all of them encompassing the equivalent region to that of bank vole rec-PrP, which spans from residues 23 to 231 (208 amino acids)], and the purity and correct folding, as evidenced by the absence of detectable oligomers or proteolytic fragments. Both, protein quality and concentration are relevant factors that affect misfolding proneness, although, as shown in Supplementary Fig. 2, within a certain range of rec-PrP concentration in the PMSA substrate (from 0.1 μg/μl to 0.025 μg/μl), results are similar.

### Spontaneous misfolding of rec-PrP by PMSA, assessment of misfolding proneness, and generation of hundreds of recombinant prions

After verifying the protein concentration and quality, each substrate underwent an in vitro procedure able to induce spontaneous misfolding of rec-PrP into bona fide prions, originally developed using bank vole rec-PrP[26]. Notably, PMSA consistently and rapidly facilitated the misfolding of this rec-PrP in less than 24 h, making it our gold standard in terms of misfolding proneness, not only because its behavior in vitro, but also because its well-documented susceptibility to both sporadic and induced prion diseases in vivo[26,27]. In order to establish a comprehensive ranking of the spontaneous misfolding propensity among the 382 proteins tested, four serial 24 h PMSA rounds (R01–R04) were performed with each substrate, using 4 replicates (samples a–d) complemented with 1 mm diameter glass beads and other 4 replicates using 0.1 mm diameter glass beads. Given the influence of the glass surface on promoting spontaneous misfolding of rec-PrP in vitro, the latter set of replicates poses more

restrictive conditions and enables a finer classification of rec-PrP with similar misfolding propensity[26]. Using the number of positive replicates for protease-resistant misfolded rec-PrP (rec-PrP^res) in each round and under varying conditions, we devised a formula to calculate the spontaneous misfolding proneness for each PrP variant. In Fig. 1, we show a schematic representation of the results for nine out of the 382 proteins tested, including examples of proteins with maximum (100 score) and minimum (0 score) misfolding proneness. The first, exhibiting 100% positive replicates for rec-PrP^res in the first PMSA round, as detected through proteinase K digestion, electrophoresis, and total protein staining (indicated in green), received a score of 100, equivalent to our benchmark, bank vole PrP. In contrast, proteins for which rec-PrP^res could not be detected in any replicate and PMSA rounds (shown in red), indicating an inability to misfold at least in absence of pre-formed seeds, resulted in a score of 0. It is worth mentioning that despite the high reproducibility of the method was confirmed before using bank vole rec-PrP[26], all the rec-PrP that resulted in a misfolding score of 0 were subjected to an additional repetition of four serial PMSA rounds to confirm their incapacity for spontaneous misfolding. In the rare cases in which some tube turned out to be positive in the second repetition, the results of the latter were used to assign the final misfolding score. Intermediate scenarios are also illustrated, with scores determined by the number of positive replicate tubes in each round and under varying conditions. The large volumes and amounts of rec-PrP^res generated in PMSA allow the use of total protein staining, facilitating the detection of misfolded proteins from multiple species and avoiding the necessity of performing Western blots and associated complications regarding antibody specificity. This information allowed ranking the misfolding propensity of 382 rec-PrP variants, detailed in Supplementary Table 2. The list spans from those exhibiting the highest to lowest misfolding scores, with variants sharing identical scores arranged alphabetically based on their species' binomial nomenclature. Together with the misfolding score, we have included the order in which each species belongs. Interestingly, this classification reveals an absence of correlation of specific taxonomic orders with misfolding propensity, as happens also with the amino acid differences compared to bank vole PrP. This underscores the idea that sequence similarity has a seemingly minor influence on spontaneous misfolding propensity, and points towards a major influence of specific changes rather than the mere quantity of variations. Additionally, using AlphaFold and Rosetta[28,29], we calculated the relative thermodynamic stability of selected globular protein isoforms with respect to bank vole PrP. No correlation was observed between the predicted thermodynamic stability and the propensity to misfold ($R^2 = 0.034$), suggesting that the misfolding proneness is independent of the thermodynamic stability of the globular isoform. Supplementary Fig. 3 provides further insights, showing the distribution of rec-PrP variant scores as percentiles (Supplementary Fig. 3A), and plots demonstrating the lack of correlation between the misfolding score and amino acid disparities relative to bank vole PrP (Supplementary Fig. 3B), as well as with the stability of the globular isoform (Supplementary Fig. 3C).

### Generation of a variable number of biochemically distinguishable conformers

As previously observed for bank vole rec-PrP, where multiple conformers were detected due to differences in the proteolytic fragments following proteinase K digestion, electrophoresis, and total protein staining, resulting in distinct prion strains[26], we noted the presence of a variable number of biochemically distinguishable rec-PrP^res conformers among the distinct replicates for more than 130 species. Although biochemically indistinguishable rec-PrP^res conformers can potentially arise into different strains upon in vivo inoculation, the identification of distinct electrophoretic mobility patterns serves as a clear indicator of potential structural differences that might represent

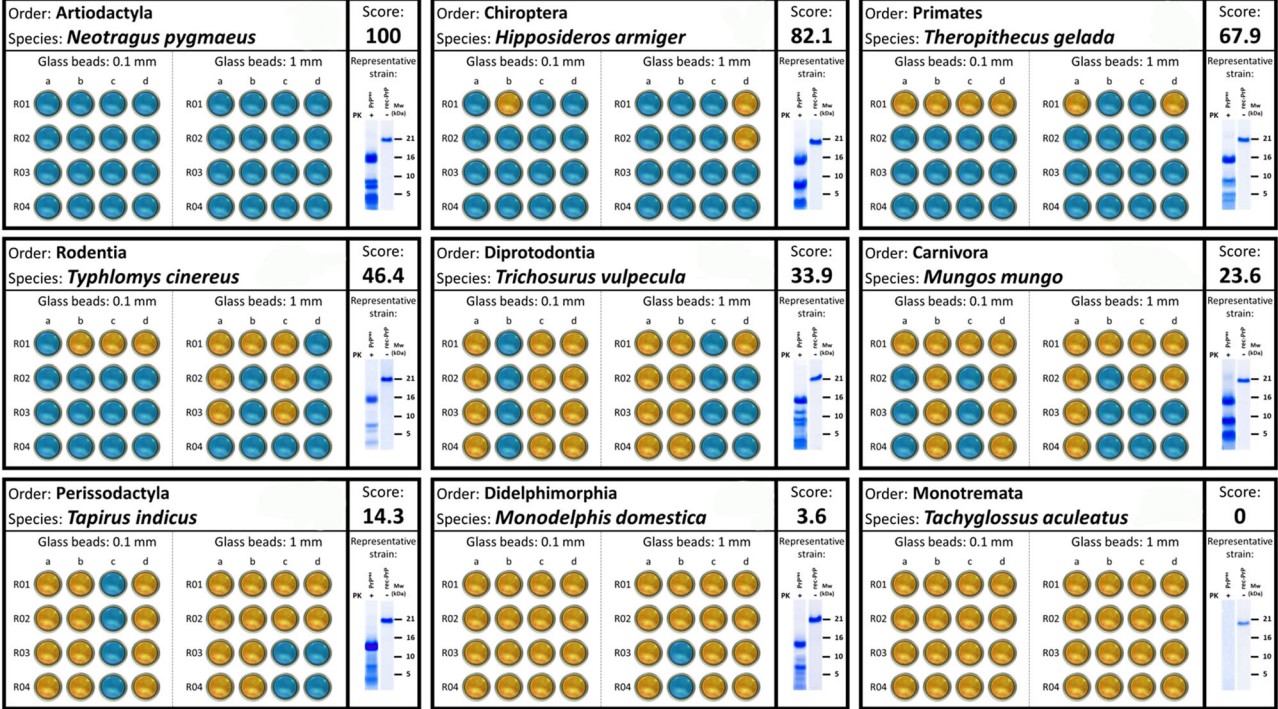

**Fig. 1 | Schematic representation of PMSA results for nine rec-PrP variants performed to comparatively assess their misfolding capacity and the determination of their misfolding proneness score.** Results from a representative sample of rec-PrP variants from 9 different species from various orders are shown. These variants exhibit misfolding proneness scores ranging from 0 to 100, as exemplified within the context of the 382 PMSA assays conducted. In addition to indicating the order, binomial names of the species, and their respective misfolding scores, the figure also shows the results obtained from the four serial rounds of PMSA (R01–R04) with each set of samples. These sets include those complemented with 1 mm glass beads or 0.1 mm glass beads, each having four replicates, from *a* to *d*. Blue dots in each panel indicate positive rec-PrP^res results following PMSA (for more information on the rec-PrP^res positivity criteria see Supplementary Methods). Conversely, orange dots denote negative PMSA outcomes for rec-PrP^res. Considering the number of rec-PrP^res positive samples in each round and the set of tubes, misfolding proneness scores (Score) were calculated for each species. In this representative sample, scores range from 100 (all replicates from both sets of samples positive from the first PMSA round) to 0 (all replicates negative even after the fourth serial PMSA round). Below the scores, the electrophoretic mobility pattern of a representative rec-PrP^res is included for each species after PK digestion, along with the undigested rec-PrP from the corresponding substrate. For those species unable to misfold after four serial PMSA rounds, a blank gel corresponding to the analysis of one of the replicates after the last round is shown. For species with multiple distinguishable conformers in their electrophoretic mobility patterns, the most frequent one was selected. For each species, the assay was pre-formed once (n = 1) using 2 sets of 4 replicate tubes, and for those rec-PrP variants with all replicates negative in all four PMSA rounds, the assay was performed twice (n = 2) confirming negative results. PK: Proteinase K, MW: Molecular weight marker. All the Source Data used for this figure is publicly available in Zenodo repository (10.5281/zenodo.10579518).

different strains, as we already demonstrated for several bank vole rec-PrP^res conformers[26], also included in the final data compilation. Consequently, we systematically examined all PMSA products to identify patterns suggesting possible strain variability, which will be further analyzed in the future through bioassays to confirm the potential strain variability obtained here. Figure 2 shows six different examples of species exhibiting a range from at least a single conformer to those with up to at least six, which represent the highest variability found for a single species. It is noteworthy that the number of conformers found for each species, that will be included in the final compilation of data as explained below, seems unrelated to the total number of rec-PrP^res positive replicates, given that, as shown in the Fig. 2, we found species with 8 out of 8 positive samples displaying a single rec-PrP^res electrophoretic pattern, whereas others, with just a few positive replicates showed higher variability. Far from being a rare occurrence, almost 35% of the PrP variants tested showed more than one conformer, with 72 variants with at least 2 distinguishable rec-PrP^res patterns, 34 with at least 3, 24 species with 4 conformers, 3 with at least 5 and two variants showing the maximum diversity observed with 6 potential strains. As shown in Supplementary Table 2, we could not detect any correlation between conformer diversity and PrP sequence, given that the 137 PrP variants displaying multiple conformers belong to different orders and do not have noticeable common elements in their sequence. This observation, based on visual inspection, does not reveal any discernible patterns or statistical trends. Moreover, as shown in Supplementary Table 2, we could not detect any correlation between the number of conformers and specific mutations in the PrP sequence, indicating that the ability to form multiple conformers is a general trait of PrPs and is not confined to a specific order.

## Infectivity of spontaneously misfolded rec-PrP^res confirms their bona fide prion nature

Apart from providing insight into the misfolding propensity of hundreds of PrP variants, the method presented herein holds the potential to provide recombinant prions from dozens of species for which prion diseases have never been reported. This contributes with models for unexplored TSEs. To assess whether the rec-PrP^res generated by PMSA indeed behave as bona fide prions, it was imperative to demonstrate their ability to induce a TSE in model organisms. Due to the existence of interspecific transmission barriers, the ideal assessment would evaluate the infectious capacity of each recombinant prion by inoculating it into a model expressing homologous PrP^C. However, the large number of distinct rec-PrP used and the exotic nature of many of the mammal species make this approach challenging at best. Therefore, to demonstrate the bona fide prion nature of the rec-PrP^res obtained spontaneously, we decided to use transgenic mice expressing bank

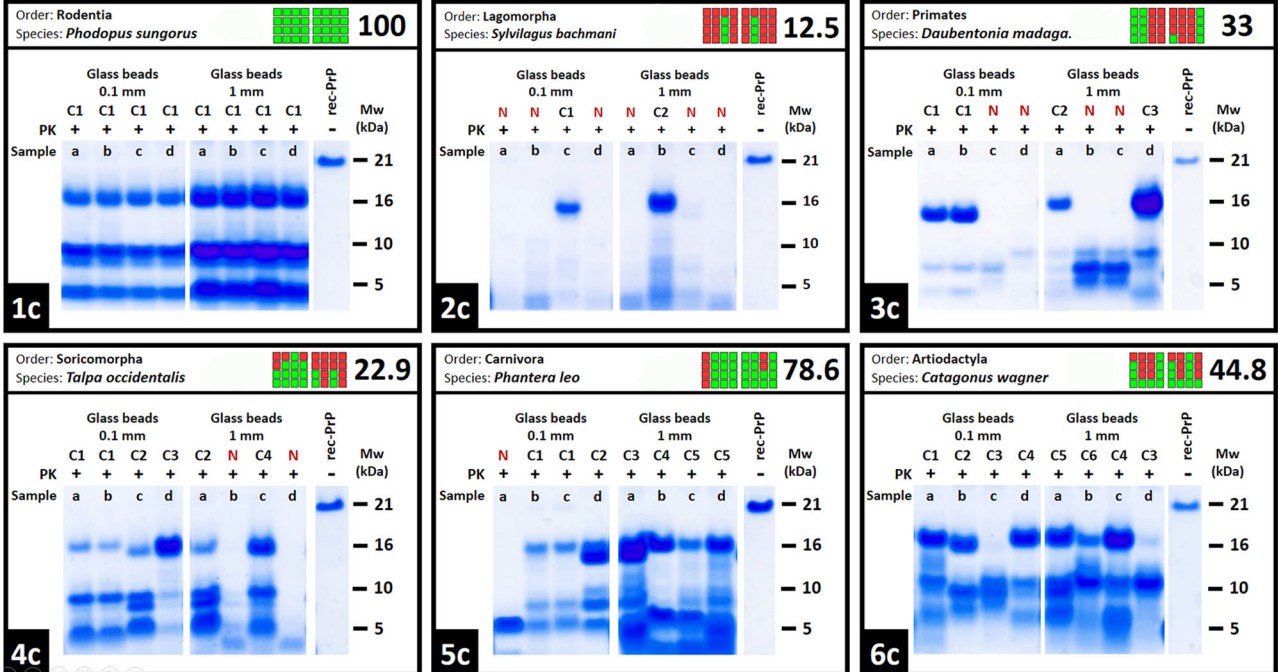

**Fig. 2 | Representative sample of biochemically distinguishable conformer variability evidenced among spontaneously misfolded rec-PrP variants.** Based on the detection of different proteolytic fragments following PK digestion, electrophoresis, and total protein staining, those rec-PrPres showing distinguishable electrophoretic mobility patterns were considered different conformers, although we acknowledge the possibility that rec-PrPres with indistinguishable patterns may also exhibit slight structural differences. To exemplify the fact that spontaneous misfolding in PMSA can give rise to multiple biochemically distinguishable conformers, with notable differences between rec-PrP variants, we show here the results from the fourth serial PMSA round involving six distinct species, a representative sample of the conformer variability observed throughout the study. This variability ranges from species that, despite having 8 positive replicates, all misfold into a single distinguishable conformer (Conformer 1 or C1, as shown in panel 1c), to others that, with just two positive replicates (N stands for negative) yielded two

distinguishable conformers (panel 2c), with all intermediate possibilities (panels 3c, 4c, and 5c) up to the detection of six conformers (panel 6c), representing the maximum variability observed in this study. Each panel includes the undigested rec-PrP from the substrate, as well as a graphical summary of the PMSA results (colored squares located in the upper part of each panel, with each horizontal square representing a PMSA replicate and vertical squares representing serial rounds; green for rec-PrPres positive and red for rec-PrPres negative samples). As shown through these examples, highest misfolding scores, and thus, more rec-PrPres positive replicates, do not necessarily lead to a greater number of distinguishable conformers. This suggests that the conformational variability arising from misfolding is independent from misfolding proneness. N: negative replicate, C_n: conformer number, PK: Proteinase K, MW Molecular weight marker. All the Source Data used for this figure is publicly available in Zenodo repository (10.5281/zenodo.10579518).

vole PrPC (TgVole), a highly susceptible species deemed as the universal acceptor of prions. Furthermore, we carried out two approaches: first, the capacity of the rec-PrPres to induce misfolding of brain-derived PrPC was evaluated through PMCA, using TgVole brain homogenate as substrate and the PMSA products as seeds. Once we confirmed that several of the rec-PrPres from distinct species were able to misfold brain-derived PrPC in vitro, an indicator of potential infectivity in vivo, the recombinant products were inoculated intracerebrally into TgVole mice. Figure 3 shows some examples of the infectivity studies done for gerbil, bat, rabbit, and mink rec-PrPres, four species belonging to four different orders. For PMCA, the rec-PrPres were partially purified through density gradient ultracentrifugation, which resulted in most cases in the separation of the rec-PrPres in a visible halo that was then used as seed. This purification step before PMCA, apart from concentrating the rec-PrPres from the sample potentially reducing the number of rounds needed for misfolding of brain PrPC, contributed to reduce the concentration of dextran sulfate from the seed, avoiding interference and artifacts observed in TgVole brain homogenate-based PMCA in the presence of high concentrations of dextran sulfate. These purified and concentrated products were used as seeds at 1:100 dilution for the first PMCA round of 24 h, up to three serial rounds were performed, second and third round seeded with a 1:10 dilution of the PMCA product from the previous round. On the contrary, for bioassay, PMSA products, simply diluted 1:10 in PBS were directly inoculated intracerebrally in TgVole animals or models

expressing homologous PrPC to the recombinant seed when available. Both PMCA and in vivo inoculations in TgVole confirmed the bona fide nature of a representative sample of the recombinant prions obtained by PMSA, given the three-banded PrPSc signal observed after PK digestion and western blotting of PMCA products and brains from inoculated animals. These prions exhibited the hallmarks of other TSEs with incubation periods ranging from 129 to 243 days post-inoculation (dpi). Although still ongoing for several of the rec-PrPres generated, to date, apart from the 4 completed assays shown in Fig. 3, infectivity studies have been initiated for 29 rec-PrPres from species belonging to 13 different orders either in vitro, in vivo or both (see Supplementary Table 3 containing information of currently ongoing assays). In addition, as these are completed, infectivity of other rec-PrPres will be evaluated, including the results for each variant in the final data compilation in their individual misfolding files.

## "Gotta catch 'em all" – The "PrPdex" as a comprehensive mammalian PrP variant encyclopedia

The large amount of data collected on the spontaneous misfolding proneness of hundreds of recombinant PrP variants prompted us to create an accessible and comprehensive database (PrPdex). PrPdex features individual files for each PrP variant we have studied, currently totaling 382 files, representing 725 wild-type PrP (all prion proteins known until now), also available as a Supplementary Data 1 with this article. These files include key details such as the common and

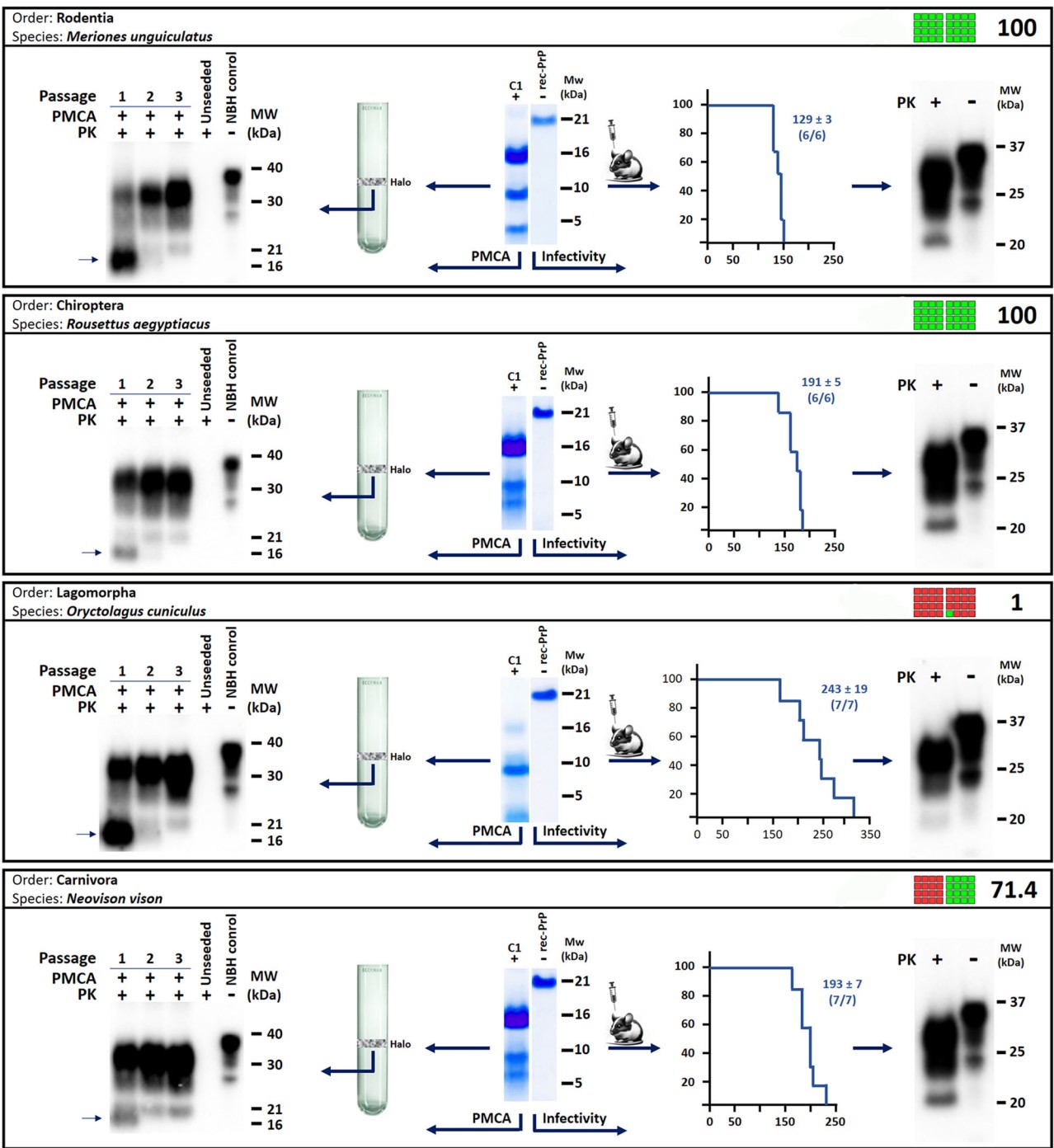

**Fig. 3 | Workflow schematic and exemplifying results for determining infectivity of rec-PrP^res generated by PMSA.** To illustrate the procedure employed to assess the bona fide prion nature of the rec-PrP^res obtained in PMSA, four examples are shown, carried out with the rec-PrP^res of gerbil, Egyptian fruit bat, rabbit, and mink, all belonging to distinct orders and exhibit varying misfolding scores. In each case, a rec-PrP^res conformer (C1) was centrally displayed in the middle of each panel alongside the respective undigested substrate control as observed after PK digestion. The left part of each panel showcases the partial purification of the rec-PrP^res through ultracentrifugation and how the purified protein halo was used as seed in PMCA diluted 1:100 in TgVole brain homogenates used as substrate. A Western blot depicting the outcomes of the three serial PMCA rounds (the first at 1:100 dilution and the rest at 1:10) performed with each seed is shown. These include an unseeded tube as control for spontaneous misfolding or cross-contamination. For each rec-PrP^res tested, two independent PMCA experiments were performed using two replicates in each experiment. As can be seen in the Western blots, all rec-PrP^res tested were able to induce misfolding of bank vole PrP^C, strongly indicating their potential infectivity in vivo. Notably, the rec-PrP^res seed is visible after the first PMCA round, indicated by a black arrow. The right part of each panel details the assessment of infectivity for these PMSA products through bioassay in TgVole. Kaplan–Meier curves depict the survival times with the mean incubation period expressed as days post-inoculation (dpi ± SEM) and the number of animals in each group succumbing to disease indicated below (# of sick animals/# of animals inoculated, being in all cases $n = 6$ to 7 animals). Additionally, the biochemical analysis of the brains of the inoculated animals is presented. By Western blot using Sha31 antibody (1:4000 dilution), all analyzed animals showed PrP^Sc, confirming the infectious nature of the recombinant prions form these four different species. C_n: selected conformer number; PK: Proteinase K, MW: Molecular weight marker. All the Source Data used for this figure is publicly available in Zenodo repository (10.5281/zenodo.10579518).

binomial names and a picture of the species, the GenBank accession number, other mammalian species sharing the same sequence (excluding differences in the first 90 amino acids, the flexible N-terminus of the PrP), the misfolding proneness score determined by PMSA, and its percentile ranking relative to other analyzed species (Supplementary Fig. 4). Additionally, we have included data on the number of species within the same order and family whose rec-PrP is able to misfold spontaneously. To facilitate comparisons with the bank vole PrP, used as the gold standard given its high spontaneous misfolding capacity, each file displays the amino acid sequence of the species (from amino acid 100 to C-terminus). Accompanying this sequence there is a visual representation of the main secondary structural motifs, with specific focus on residues that diverge between species. Pertinent information related to misfolding, such as the β2-α2 loop sequence and the presence of indels and polymorphic variants, is thoughtfully provided. The PMSA results contributing to the misfolding score are also featured in each file, offering insights into the number of rec-PrP[res] positive replicates for each PMSA round, as well as the electrophoretic mobility patterns and the count of distinct conformers detected in each case. A summary of the infectivity results, encompassing both in vitro and in vivo outcomes (in TgVole or homologous species when available), is also included. Further enriching this resource are the AlphaFold-predicted globular PrP structures (available for download and displayed as interactive 3D protein structures) alongside their corresponding predicted local distance difference test (pLDDT, average of per-residue AlphaFold prediction confidence, ranging from 0 to 100), and the calculated relative thermodynamic stability values complemented by experimentally determined melting temperature ($T_m$) values, when available.

This dynamic and continually expanding encyclopedia can be accessed at PrPdex. Its mission is to serve as an all-encompassing repository of prion protein sequences, providing a one-stop source for all relevant information about each analyzed PrP variant. We anticipate ongoing growth with the inclusion of polymorphic variants and newly sequenced species, making PrPdex a valuable resource for researchers investigating protein misfolding-related disorders.

## Discussion

Although the "protein only" hypothesis was postulated more than 40 years ago[1] and proved more than a decade back[23], it is hitherto unknown why TSEs have only been reported in a handful of mammalian species or how diverse PrP sequences lead to differential susceptibility to prion infections. Despite the high evolutionary conservation of PrP in mammals and its structural conservation in other classes[30], remarkable differences have been reported among mammalian species and polymorphic variants regarding their susceptibility to TSEs[31]. However, which elements from the sequence determine the capacity of a specific PrP to undergo misfolding into the pathogenic isoform remain elusive, precluding the prediction of new potential cross-species transmission events or the most probable future outbreaks among others. Previous studies, often constrained by the limited number of PrP variants analyzed, mainly relied on exogenously induced infections in a few select animal models, or solely relied on data from naturally occurring TSE infections or laboratory models[32–34]. Nonetheless, all these approaches failed to pinpoint the critical sequence elements that dictate misfolding susceptibility. A systematic approach, performing a high number of random changes for a given PrP sequence could provide further information, beyond what has been described in nature. However, this approach would inevitably lead to the design of proteins unable to fold into a stable globular structure, and therefore, useless in terms of analyzing misfolding determinants. Consequently, we decided to tackle the study using as many prion protein variants as possible, taking advantage of the diversity shaped by the evolution within the Mammalia class. In addition, given that the induced misfolding of a PrP depends on a yet

uncharacterized structural compatibility between exogenous PrP[Sc] and the endogenous PrP[C][31], which could introduce some bias into the assessment of PrP[C] misfolding propensity, we chose to assess the spontaneous misfolding proneness as a more accurate measurement of conversion susceptibility. In particular, we devised a method to assess misfolding propensity of those PrP variants able to give raise to protease-resistant misfolded forms specifically, due to the screening method used based on proteinase K digestion and the criteria established. This consisted of considering rec-PrP[res] positive, only those misfolded products in which PK-resistant fragment corresponding to the amyloid fiber core expanding from residues around 90 to the C-terminal end of the protein was detectable after electrophoresis and total protein staining with the naked eye, which in the case of the rec-PrP[res] shown in this work is a proteolytic fragment of approximately 16 kDa. Given this approach, we cannot exclude the possibility that some of the species considered unable to misfold here, could have been misfolded into protease-sensitive forms, since this kind of prions, although rarely, have been also describe in nature[35].

Considering amino acids 90 to the C-terminus end of the prion protein as the most influential on misfolding tendency–PrP variants of approximately 141 amino acids–the variety of sequences found in distinct mammals can be illustrated by the existence of alternative residues in 102 out of those 141 positions, some species differing in up to 30 amino acids. Alterations in the amino-terminal extreme of the protein, despite may be important in some context for PrP misfolding[36] do not seemingly influence the misfolding capacity of rec-PrP in the PMSA system based on our experience with proteins with various tags or modifications in this region. However, we cannot rule out completely that for some species that were grouped together taking into consideration differences only from amino acids 90 to C-ter, there could be some difference in the spontaneous misfolding score assigned here. In any case, and despite some limitations of the method, the diversity of PrP sequences analyzed, that we roughly estimate to cover about 50% of the variability existing in mammalian species (considering the similarity of sequences within each genre, the total number of species in our collection, and the approximately 5400 total mammal species), provide a basis for the analysis of the effect of PrP sequence elements in their spontaneous misfolding proneness.

Furthermore, the discovery of prion-like proteins, understood as proteins able to switch conformation from a functional to an amyloidogenic and transmissible isoform, across different organisms spanning distinct domains has offered another important source of data to try to understand the molecular determinants allowing such a particular conformational change. Mostly based on yeast prions[37], the so-called prion domains (PrD) were identified, short glutamine and asparagine-rich regions devoid of hydrophobic or charged residues, that differ significantly from the standard elements found in canonical amyloid-forming proteins. Nonetheless, and despite the fact that numerous bioinformatics tools developed based on this knowledge have allowed the identification of several previously unknown prion-like proteins[38], the amino acid determinants driving the misfolding of the prion protein remain elusive. Therefore, an experimental approach, such as the one presented in this manuscript seems necessary to understand the complex PrP misfolding event on a molecular level. In fact, the possibility of developing dedicated bioinformatic tools able to predict the behavior of a prion protein solely based on its sequence could be within reach thanks to the data obtained from this high-throughput analysis of all imaginable PrP variants. As an example, the correlation analyses presented here indicate that specific changes in the context of a PrP sequence may exert a stronger influence on spontaneous misfolding proneness than the total number of variations with respect to another PrP. Similarly, our results suggest that the misfolding propensity is independent of the thermodynamic stability of the globular rec-PrP isoform. Further conclusions, such as the determination of specific residues, hot spots or regions that could

be most influential to the spontaneous misfolding proneness of PrP may be theoretically achievable based on data like the one presented here. However, due to the limitations of human visual analysis in handling large and complex datasets such as the one presented here, at this stage, we were unable to pinpoint any position or region that could determine misfolding ability or incapability regardless of the rest of the PrP sequence or find any universal rule modulating spontaneous prion misfolding. We believe that finding general rules applicable to spontaneous PrP misfolding would require an in-depth analysis of the results through bioinformatics and machine learning approaches, what in turn may require larger datasets and dedicated studies. For this reason, we are currently working on increasing the number of PrP variants for analysis, including polymorphic variants described in some species that are, in addition, partially related to prion disease susceptibility data in vivo; and ad hoc designed artificial PrP variants (point mutants, indel mutants, chimeric constructs) specifically aimed to address if there is some critical position or domain determining spontaneous misfolding capacity regardless of neighboring or even distant regions. Furthermore, the idea of making the method and the current dataset available to the prion research community intends to accelerate this kind of studies, given that understanding such a complex phenomenon will likely require distinct approaches and a joint effort.

The PMSA method presented here facilitates the spontaneous generation of bona fide infectious prions from virtually any PrP capable to misfold into its pathogenic counterpart. This method not only sheds light on their misfolding proneness but also yields multiple conformers for a single PrP sequence, mimicking the strain diversity observed in nature. An essential element enabling efficient spontaneous misfolding in our system is dextran sulfate, a polyanionic cofactor chosen among the several distinct cofactors shown to enhance PrP misfolding, at least in vitro. This cofactor functions in a potentially similar manner to others previously described, such as RNA[16] or heparan sulfate[39]. Even though the exact role of these cofactors remains to be fully elucidated, research from our group[40] and others[41] suggests they may have some role on driving specific strain properties. Moreover, a recent report shows that certain PrP sequences may be stimulated to misfold in the presence of a particular cofactor, which may at the same time show no effect on other PrP variants[42]. In the context of our method, where dextran sulfate was selected due to its misfolding-promoting effect observed for bank vole rec-PrP, it is plausible that some PrP sequences could exhibit varying misfolding propensities in the presence of other cofactors. In this regard, it is important to highlight that some of the results shown, such as the incapacity of Syrian hamster PrP to misfold, conflict with previous studies in which they have been used as animal models of prion disease, being susceptible to various prion strains[43]. These discrepancies may respond to their specific behavior in presence of dextran sulfate, which may promote misfolding of PrP into a specific type of conformer, somehow different to known prion strains, and impossible to adopt for some PrP variants. This cofactor specificity, despite introducing a bias in the method developed, could be however useful to further explore strain-cofactor relationship and indicates the potential of the PMSA system for the study of strain determinants. Furthermore, the emergence of multiple conformers, potentially representing distinct strains, within PMSA is remarkable although not entirely unexpected, given prior results obtained with bank vole rec-PrP[26]. This variability could suggest that the dextran sulfate may not be very restrictive in driving specific misfolded PrP conformations, further bolstering the presented findings. However, the formation of variable numbers of conformers could be an intrinsic property of each PrP variant. If, as previously proposed, the spontaneous formation of misfolded propagation nuclei is an infrequent occurrence, followed by a very rapid propagation of the initial conformer, the formation of distinct misfolded isoforms could respond to stochastic misfolding

events. This, in turn, could lead to a larger variety of potential strains than those detected through repeated procedures. In any case, this analysis of 382 PrP variants indicates that some PrP sequences could be more prone to adopt slightly distinct conformations upon misfolding than others, mirroring the diversity of strains observed in different mammal species in nature.

The wealth of data in this catalog provides numerous opportunities for addressing pivotal questions in the field. These questions range from understanding the features of a PrP sequence that promote misfolding into its pathogenic counterpart to deciphering infectivity determinants, specific regions or sequences driving transmission barriers among species or polymorphic variants, and even the structural influence of specific amino acids. From a practical perspective, this resource offers an abundance of models for structural studies, designing advanced substrates for ultrasensitive prion detection, developing animal models with enhanced misfolding susceptibility, engineering prions with specific traits such as rapidness or determined host range to speed up preclinical studies, or designing dominant negative proteins for therapeutic applications.

## Methods

### Ethical statement

All experiments adhered to the guidelines included in the Spanish law "Real Decreto 53/2013 de 1 de febrero" on protection of animals used for experimentation and other scientific purposes, which is based on the European Directive 2010/63/UE on Laboratory Animal Protection. The project was approved by the Ethical Committees on Animal Welfare [project codes assigned by the Ethical Committees P-CBG-CBBA-0519 (CIC bioGUNE), NEIKER-OEBA-2021-003 (Neiker) and project 11926 (CReSA)] and performed under their supervision.

### Mouse work

All bioassay or prion infectivity studies described in this article were carried out using TgVole 1× mice [FVB/N-Tg(Prnp-BVole109I) C594PRC/J] ($n = 5–7$) groups. Female TgVole 1× animals [age, sex, and randomization codes are detailed in the Source Data, publicly available in Zenodo (10.5281/zenodo.10579518)[44]] were used in all cases due to their lower aggressiveness when housed together for long periods of time, that were blindly and randomly allocated in experimental groups, through an internal coding system. This ensured equitable distribution of inherent variabilities. Additionally, to minimize biases, blind allocation was employed, with the personnel assessing clinical signs unaware of specific group assignments. Mice were kept in a controlled environment at a room temperature of 22 °C, 12 h light-darkness cycle, and 60% relative humidity. They were fed *ad libitum* and examined at least twice a week, upon development of neurological signs of disease monitoring was increased to daily observation. The clinical signs monitored included kyphosis, gait abnormalities, altered coat state, depressed mental state, flattened back, eye discharge, hyperactivity, loss of body condition, and incontinence. Clinically affected animals with two or more severe signs or invalidating motor disturbances were euthanized before neurological impairment compromised their welfare, by exposure to a rising concentration of carbon dioxide or alternatively, by cervical dislocation. Results are expressed as mean incubation period in days post-inoculation (dpi), of all animals inoculated in each group in which prion disease could be confirmed either by western blot or immunohistochemistry, providing all the standard error of the mean (SEM) for each experimental group. Additionally, attack rate was calculated considering the number of animals with confirmed prion disease and the total number of mice inoculated in each group. As control group, TgVole 1× mice inoculated with non-fibrillary bank vole rec-PrP, published in[45], were considered. All TgVole 1× mice were bred at CIC bioGUNE (Spain) and inoculated at Neiker−Basque Institute for Agricultural Research and Development and IRTA-Centre de Recerca en Sanitat Animal (CReSA).

## Recombinant prion protein expression and purification

Expression and purification of all 382 recombinant PrP used during the study (see list of all sequences, publicly available in Genbank, with their corresponding accession numbers in Supplementary Table 1) was carried out as described previously[26]. Although all the details of the procedure, including tips and troubleshooting, are described thoroughly in Supplementary Methods section, briefly, all rec-PrP sequences (from amino acids 23 to 231 in bank vole PrP numbering, devoid of the N-ter signal peptide), either obtained from genomic DNA of the species through PCR or by synthesis, were cloned into pOPIN E expression vector, developed by Oxford Protein Production Facility UK (OPPF). After transforming them in *E. coli* Rosetta™ (DE3) competent cells (EMD Millipore), they were grown in ampicillin-containing LB broth and recombinant protein expression induced using Isopropyl β-D-1-thiogalactopyranoside (IPTG) (Gold biotechnology). Purification was done through immobilized metal affinity chromatography using His-trap columns (HisTrap FF crude 5 ml, GE Healthcare Amersham) taking advantage of the histidine-rich octapeptide repeat (OR) from all PrP sequences. After solubilizing inclusion bodies containing most of the rec-PrP, the clarified cell extract was loaded into the column, washed, and eluted (in elution buffer, 20 mM Tris-HCl, 500 mM NaCl, 500 mM imidazole and 2 M guanidine-HCl, pH 8). After elution, guanidine-HCl concentration was increased up to 6 M for long-term storage at −80 °C and then, protein concentration was adjusted to 25 mg/ml using centrifugal filter units (Amicon® Ultra-15 PLGC Ultracel-PL 10 KDa, Millipore) and performing measurements of absorbance at 280 nm (Nanodrop). Finally, protein amount and purity were evaluated by total protein staining (BlueSafe, Nzytech) after methanol precipitation and electrophoresis in SDS-PAGE gels (Bio-Rad).

## Substrate preparation for Protein Misfolding Shaking Amplification

Preparation of rec-PrP based PMSA substrates for spontaneous rec-PrP^res formation, also described in more detail in Supplementary material, was done as follows. The purified rec-PrP stored at −80 °C was diluted 1:5 in phosphate buffered saline (PBS, Hyclone) and dialyzed for re-folding against PBS at 1:2,000 ratio for 1 h at room temperature. The dialyzed protein was centrifuged at $19,000 \times g$ for 15 min at 4 °C to eliminate amorphous aggregates and the supernatant, containing only soluble rec-PrP, was used for substrate preparation. The dialyzed proteins, with a concentration of around 20 μM given similar yields upon re-folding, were diluted in conversion buffer (CB)[26] 1:9, and dextran sulfate sodium salt from *Leuconostoc spp.* with molecular weights ranging from 6500 to 10,000 (Sigma–Aldrich) was added to a final concentration of 0.5% (w/v). All substrates were aliquoted and stored at −80 °C until required.

## Protein Misfolding Shaking Amplification

To induce spontaneous misfolding of the rec-PrP variants and rank their misfolding proneness, a specific Protein Misfolding Shaking Amplification (PMSA) methodology was designed, further described in Supplementary Methods. Briefly, all different substrates were disposed in eight 2 ml tubes with conical bottom and screw cap (Fisherbrand), four of them complemented with 100 mg of 1 mm diameter acid-washed glass beads (BioSpec Products, Inc.), and the other four with 100 mg of 0.1 mm diameter acid-washed glass beads (BioSpec Products, Inc.). All 8 tubes for each one of the 382 substrates prepared were then submitted to PMSA at 39 °C using either a Thermomixer (Eppendorf) or a Digital shaking Drybath (Thermo Scientific) with internal temperature control and shaking at 700 rpm continuously for 24 h. Four 24 h serial PMSA rounds were performed for each substrate, diluting 1:10 the PMSA product from the previous round into a new set of 8 tubes with freshly thawed substrate and their corresponding beads. Those rec-PrP variants with all replicates negative for rec-PrP^res

in all four PMSA rounds, were performed twice to confirm the negative results.

## rec-PrP^res detection

All PMSA products from the four serial rounds were transferred from the reaction tubes to 1.5 ml Eppendorf tubes and digested by adding proteinase K (PK) (Roche) at 25 μg/ml for 1 h at 42 °C in an oven (Nahita). Following digestion, samples were centrifuged at $19,000 \times g$ at 4 °C for 15 min, the supernatant discarded, and the pellets, containing only PK-resistant and insoluble rec-PrP^res, were resuspended and washed with at least 700 μl of PBS (Fisher Bioreagents). Then, samples were centrifuged for another 5 min at $19,000 \times g$ and 4 °C, and the pellets were resuspended in 15 μl of loading buffer 4× (NuPage LDS, Invitrogen), previously diluted to 1× in PBS. PK-resistant PrP detection was done through electrophoresis and total protein staining. PK-digested samples were boiled for 10 min at 100 °C and loaded onto 4–12% acrylamide gels (NuPAGE Midi gel, Invitrogen Life Technologies). The electrophoresis was run for 1 h and 20 min (10 min at 70 V, 10 min at 110 V, and 1 h at 150 V) and stained with BlueSafe (NZYTech) for 1 h at room temperature. To minimize variations from comparing samples run in different gels, the same amount of PMSA product was processed in all cases and the different process times were strictly controlled, especially the staining time. Additionally, different PMSA substrates (containing similar rec-PrP concentrations) were run along the digested PMSA products in most gels, serving as internal controls of the staining reagent to avoid dramatic differences between gels. All gels were scanned using the iBright™ CL750 Imaging System (Invitrogen) and analyzed using AlphaView software (version 3.4.0, ProteinSimple).

## Scoring of spontaneous misfolding proneness

To rank the spontaneous misfolding proneness of the 382 distinct rec-PrP analyzed in PMSA, the following formula was designed, considering the number of rec-PrP^res positive tubes determined by PK digestion electrophoresis and total protein staining in each serial PMSA round (where $n$ is the number of rec-PrP^res positive tubes and the sub-indexed number indicates the PMSA round), and the type of glass beads of the tube (either 0.1 mm diameter or 1 mm diameter glass beads). The assessment of rec-PrP^res positive PMSA products (depicted as green dots or squares in Figs. 1, 2, and 3 and in the PrPdex files) was based on the detection after proteinase K digestion, electrophoresis and total protein staining with the naked eye of a proteolytic fragment corresponding to the equivalent digested product in brain-derived prions, that is the amyloid fiber core expanding from residues around 90 to the C-terminal end of the protein, which in the case of most of the rec-PrP^res shown in this work is of around 16 kDa in size.

$$S_{0.1mm} = 0.4*(n_1*10 + n_2*6 + n_3*3 + n_4*1) \tag{1}$$

$$S_{1mm} = n_1*10 + n_2*6 + n_3*3 + n_4*1 \tag{2}$$

$$S = \frac{(S_{0.1mm} + S_{1mm})}{112}*100 \tag{3}$$

The tube sets with one or the other size of beads in which rec-PrP^res was detected, received distinct values, being the misfolding score (S) assigned to 0.1 mm diameter beads lower than that assigned to 1 mm diameter beads. This distinction was made according to previous observations indicating that rec-PrP^res misfolding was favored not just by the presence of a determined glass surface but also strongly influenced by bead movement[26]. Therefore, tubes complemented with 1 mm diameter glass beads resulted more restrictive in terms of inducing spontaneous misfolding of rec-PrP, which led us to assign a higher value to these positive tubes than to those complemented with

0.1 mm. Source data available in a Dataset deposited in the Zenodo public repository (10.5281/zenodo.10579518)[44].

### Determination of infectivity of the rec-PrP$^{res}$

To check if the rec-PrP$^{res}$ generated by PMSA can induce misfolding of PrP$^C$ from brain homogenates and induce a TSE in animals, two approaches were undertaken. Given the variety of rec-PrPs and the exotic nature of many species, we decided to use transgenic mice expressing the prion protein from bank vole (*Myodes glareolus*) with the I109 polymorphism (GenBank accession number AF367624) 1-fold, generated as detailed in[45]. This animal model, highly susceptible to TSEs, presents a wide host range[27], which makes it ideal to evaluate the potential infectious capacity of the various rec-PrP$^{res}$ generated in PMSA.

To check whether the rec-PrP$^{res}$ generated by PMSA could induce misfolding of brain-derived PrP$^C$, a strong indicator of infectivity in vivo[26], we used them as seeds in PMCA with TgVole 1× brain homogenate as substrate. For that, perfused brains from TgVole 1× mice were homogenized at 10% (w/v) in CB with protease inhibitor cocktail (Roche) in glass potter pestles (Fisher Scientific), aliquoted and stored at −80 °C until required. The rec-PrP$^{res}$ were partially purified prior to their use as seeds by ultracentrifugation in density gradients. Briefly, the rec-PrP$^{res}$ from 15 ml of each PMSA product was concentrated by centrifugation at 19,000 × *g* and 4 °C for 15 min. Approximately 2 ml concentrated rec-PrP$^{res}$ were then loaded on top of a continuous sulfate cesium (Sigma−Aldrich) gradient ranging from 1 M to 1.7 M, prepared in PBS using a gradient mixer (Sigma−Aldrich), all placed in Thinwall Ultra-Clear 13.2 ml centrifuge tubes (Beckman Coulter). After ultracentrifugation, using a SW41 Ti Swinging bucket rotor (Beckman Coulter) and an Optima L-90K ultracentrifuge (Beckman Coulter), at 210,000 × *g* for 15 h at 20 °C, one or two visible protein halos could be detected in all cases. The fraction(s) presenting a visible halo were transferred to clean 5 ml Eppendorf tubes and diluted up to 5 ml with MilliQ water for washing. Upon centrifugation at 4000 × *g* for 30 min, the rec-PrP$^{res}$ containing pellets were placed into 1.5 ml Eppendorf tubes and washed with 1 ml of PBS, and submitted to centrifugation at 19,000 × *g* for 15 min, repeating the PBS resuspension and centrifugation steps at least two times. Finally, the purified fractions were resuspended in 25–75 μl of PBS and used as seed in PMCA reactions that were performed as previously described[45]. Briefly, substrate was seeded with a 1:100 dilution of the partially purified PMSA product and a 24 h PMCA was performed in an S-4000 Misonix sonicator with microplate system (Qsonica) with incubation cycles of 15 min followed by sonication pulses of 20 s at 80% power at 38 °C regulated by a circulating water bath. Up to three serial PMCA rounds were carried out, with substrates for rounds two and three seeded at 1:10 dilution with the PMCA products of the previous round under the same conditions. To avoid cross-contamination, all PMCA tubes were sealed with plastic film (Parafilm) prior to introduction in the bath sonicator to prevent accidental opening. Unseeded tubes were also included in the same PMCA round as controls for spontaneous misfolding or cross-contamination. Alternatively, for rec-PrP$^{res}$ for which brain samples containing homologous PrP$^C$ were available, these were used as PMCA substrates instead of TgVole 1× brains, prepared in the same way and performing an identical procedure. Additionally, in the case of bank vole rec-PrP$^{res}$, its potential cross-species transmissibility was evaluated using perfused C57BL/6 wild-type mouse brain homogenate as substrate.

Although the previous in vitro propagation capacity test upon brain-derived PrP$^C$ is a strong indicator of infectivity, inoculation in animal models is needed to prove the infectious, bona fide prion nature of the rec-PrP$^{res}$ generated through PMSA. For that, rec-PrP$^{res}$ containing inocula were prepared diluting 1:10 each PMSA product in sterile DPBS (Invitrogen). The protease-resistant rec-PrP amount, estimated by electrophoresis and total protein staining, was comparable in all samples. Intracerebral inoculations of several distinct PMSA products were carried out in groups of 5–7 TgVole 1× mice [FVB/N-

Tg(Prnp-BVole109I)C594PRC/J], each of them receiving 20 μl of inoculum using a sterile disposable 27-gauge hypodermic needle while under gaseous anesthesia (Isoflurane, IsoVet®, Braun). Female TgVole 1× animals [age, sex, and randomization codes are detailed in the Source Data, publicly available in Zenodo (10.5281/zenodo.10579518)[44]] were used in all cases due to their lower aggressiveness when housed together for long periods of time. They were fed *ad libitum* and examined at least twice a week, upon development of neurological signs of disease monitoring was increased to daily observation. The clinical signs monitored included kyphosis, gait abnormalities, altered coat state, depressed mental state, flattened back, eye discharge, hyperactivity, loss of body condition, and incontinence. Clinically affected animals with two or more severe signs or invalidating motor disturbances were euthanized before neurological impairment compromised their welfare, by exposure to a rising concentration of carbon dioxide or alternatively, by cervical dislocation. Survival time was calculated as the interval between inoculation and sacrifice in days (days post-inoculation, dpi). The brains from all inoculated animals were harvested and divided sagittally, fixing one half in formalin and storing the other half at −80 °C for subsequent biochemical and anatomopathological analysis.

### PrP$^{Sc}$ detection in PMCA products and brains of inoculated animals

To confirm that the rec-PrP$^{res}$ could induce misfolding of brain-derived PrP$^C$ in PMCA or if inoculated animals succumbed to a bona fide prion disease, presence of protease-resistant, disease-associated misfolded PrP (PrP$^{Sc}$) was evaluated in PMCA products and brain homogenates of deceased animals. For that, 10% brain homogenates or PMCA products were diluted 1:1 (v/v) in digestion buffer [2% (w/v) Tween-20 (Sigma−Aldrich), 2% (v/v) NP-40 (Sigma−Aldrich) and 5% (w/v) Sarkosyl (Sigma−Aldrich) in PBS] and digested with PK (Roche) at 85 μg/ml and 42 °C for 1 h with moderate shaking (450 rpm). Digestion was stopped by adding loading buffer (NuPage 4× Loading Buffer, Invitrogen) 1:3 (v/v) boiling samples for 10 min at 100 °C. Digested samples were then loaded on 4–12% acrylamide gels (NuPAGE Midi gel Invitrogen Life Technologies), and run for approximately 1 h and 20 min. Transference to PVDF membranes (Trans-Blot Turbo Transfer Pack, Bio-Rad) was done using the Trans-Blot® Turbo™ transfer system (Bio-Rad), after which the membranes were blocked by incubation in 5% non-fat milk powder for 1 h at room temperature. As primary antibody, Sha31[46] (Bertin Bioreagents, cat. No. A03213, clone Sha31, lot no. 2013) was used at 1:4000 dilution in 2% (w/v) Tween-20 (Sigma−Aldrich) with 0.1% non-fat milk powder in PBS, incubating the membranes for 1 h at room temperature. After washing the membranes with washing buffer [2% (w/v) Tween-20 in PBS] three times, peroxidase-conjugated secondary anti-mouse antibody [(m-IgGκ BP-HRP, Santa Cruz Biotechnology, cat. No. sc-516102, lot no. D1123)] antibody, diluted 1:3000 in the same buffer as the primary antibody, was added and membranes incubated for 1 h at room temperature. Finally, after washing the membranes again three times, they were developed with an enhanced chemiluminescent horseradish peroxidase substrate (West Pico Plus, Thermo Scientific), using iBright™ CL750 Imaging System (Invitrogen) for image acquisition and AlphaView (version 3.4.0, ProteinSimple) software for image analysis. Source data in the form of uncropped blots and raw data on animal inoculation and survival times are available in a Dataset deposited in the Zenodo public repository (10.5281/zenodo.10579518)[44]. Further information on the reagents used in all these procedures (company and catalog number) is available in the Supplementary Information, at the end of the Supplementary Methods.

### Computational methods, AlphaFold prediction of globular PrP structures, and calculation of relative thermodynamic stability

The globular structures of the 382 prion sequences (Supplementary Table 2) were generated using AlphaFold 2.3.2[28] using multiple

sequence alignments, omitting the flexible N-terminus (positions 1-88) and using the following options:

*--db-preset=reduced_dbs*
*--model-preset=monomer*
*--max-template-date 2050-01-01*
*--enable-gpu-relax*
*--models-to-relax all*

Since AlphaFold structure predictions are not deterministic, for each sequence 30 independent predictions of the five neural network models were run, yielding a total of 150 structures per prion sequence. Multiple sequence alignments were not recycled but generated independently for each prion sequence. All predicted structures show high confidence and fold into the correct globular structure. The top-ranked structure (predicted with the highest confidence, i.e., with the highest global pLDDT) was chosen for visualization starting from the structured β1 strand. Structures are colored according to per-residue pLDDT from 36 (blue) to 99 (red). All figures were generated with PyMOL 2.4.0[47].

The relative thermodynamic stability of selected structures with respect to a common reference (the bank vole sequence) was predicted using the methodology described in[29]. This approach relies on approximating the folding free energy $\Delta G_f$ of the protein through averaged Rosetta's[48] energies over a number of AlphaFold-predicted structures, and calculating the relative thermodynamic stability with respect to the reference as the difference in folding free energy: $\Delta\Delta G_f = \Delta G_{f,query} - \Delta G_{f,reference}$. Since the Rosetta energy is extremely sensitive to small conformational changes in the 89–128 flexible region across different AlphaFold replicas, the reference bank vole AlphaFold models were cropped to include only the 103 amino acids between M129 and S231 prior to minimization. Also, as the Rosetta energy is dependent on the number of amino acids of the structural model, meaningful comparisons can be drawn only between models comprising the same number of residues. For this reason, relative stabilities were computed only for those sequences whose AlphaFold models comprise 103 amino acids when cropped between either M129 or L130 and the C-terminus (284 sequences, see Dataset corresponding to Supplementary Fig. 3C deposited in 10.5281/zenodo.10579518[44]). Here, $\Delta G_f$ values for each sequence were computed by optimizing the geometries of the 30 AlphaFold model 1 structures with the *minimize* application in Rosetta 3.4[49], using the following options:

*run:min_type lbfgs_armijo_nonmonotone*
*run:min_tolerance 0.001*

and taking the average of the 25 lowest energies. Structures that are more or less stable than the reference show negative or positive $\Delta\Delta G_f$ values, respectively. All energy units are given in REU (Rosetta Energy Units).

### Determination of melting temperatures for distinct rec-PrP by circular dichroism

To assess if there could be some correlation between the thermal stability of the analyzed rec-PrP and their spontaneous misfolding proneness, the melting temperature ($T_m$) of several natively folded rec-PrP was experimentally determined by circular dichroism (CD). For that, rec-PrP produced and purified as explained above, were diluted 1:5 in PBS and then dialyzed at room temperature against a sodium acetate buffer [Sodium phosphate dibasic anhydrous (ACS) 10 mM, pH 5.8], reaching a 1:1,000,000 ratio, during 1 h and centrifuged at 19,000 × $g$ for 15 min at 4 °C to eliminate amorphous protein aggregates. Once dialyzed, protein concentration was adjusted to 0.03 mg/ml using a protein quantification assay (BCA assay, Thermo Scientific) and diluting the protein in the same sodium acetate buffer when necessary. All samples were placed in 5 mm quartz cuvette (Macro cell 100-QS, 5 mm; Hellma Analytics) and analyzed using a Jasco J-810 spectropolarimeter coupled to equipped with a Peltier temperature control unit, scanning at 222 nm and temperature ranging from 20 to 90 °C. Three independent measurements ($n = 3$) were performed for bank vole, sheep, and mule deer rec-PrP to assess interexperimental variability. Given the good reproducibility of the method, the thermal stability of the rest of the rec-PrP was performed once ($n = 1$). Raw data available in a Dataset deposited in the Zenodo public repository (10.5281/zenodo.10579518)[44].

For each variant, $T_m$ values were computed by fitting ellipticity values ($E$) versus temperature to a two-state (*folded/unfolded*) model. The fitting was performed through a least-squares minimization of the error between the measured ellipticity values $E$ and the simulated ellipticity values $E_{sim}$ obtained using the following equation:

$$E_{sim} = FF_{sim}(m_1 T + b_1) + (1 - FF_{sim})(m_2 T + b_2) \qquad (4)$$

where $FF_{sim}$ is the fraction of protein in the native (folded) state, T is the temperature, and the parameters $m_1$, $b_1$, $m_2$, and $b_2$ are the slopes and intercepts of the (linear) ellipticity in the native and denatured state, respectively. $FF_{sim}$ is defined as a parametric function where $\Delta H_m$ (the enthalpy of unfolding at the melting temperature) and $T_m$ are the parameters to optimize and $\Delta C_p$, the change in specific heat capacity of unfolding, is a constant. $\Delta C_p$ is estimated from the number of residues (adjusted to the size of each analyzed rec-PrP, ranging from 202 to 211 residues). The initial guess values for $\Delta H_m$ and $T_m$ were set to 200 kJ mol$^{-1}$ and 333.15 K (60 °C), respectively. Initial guesses of $m_1$, $b_1$, $m_2$, and $b_2$ were obtained from linear fitting of the first 20 ellipticity points ($m_1$, $b_1$) and the last 20 ellipticity points ($m_2$, $b_2$).

### Statistics and reproducibility

No statistical method was used to predetermine sample size for the PMSA studies, given that all available PrP sequences from mammal species were included in the study, without excluding any from the analysis. The reproducibility of the assay was evaluated in a previous study[26] during the development of the method, using bank vole rec-PrP. Additionally, for those rec-PrP with negative results, the assay was repeated twice, further confirming the reproducibility of the assay, given that with few exceptions, completely negative results were obtained again upon repetition. Specifically, only 16% of the species evaluated twice showed different results upon repetition, although variation was minimal for more than half of them, varying from a misfolding score of 0 to a score below 3, meaning the differences were found only in one or two tubes out of the 8 replicates, greater variations being only detected in 7 out of the 130 repeated assays (5.4%). Regarding inoculation experiments in TgVole models, considering the exploratory nature of the experiment, that intended to determine infectivity of potential pathogens for which there is no prior information, no formal sample size calculation could be performed. Therefore, the number of animals ($n = 5$–7) was decided based on previous one-to-one animal transmission studies[50], in which for pilot studies, groups of 5 animals have been deemed sufficient, including up to 7 animals per group to prevent losses due to intercurrent diseases. The mean of the incubation period from inoculation to development of clinical signs, as well as the standard error of the mean (SEM) are provided, all calculated using Excel 365 software. For correlation analyses, linear regression model was used providing the R-squared values for each comparison shown in Supplementary Fig. 3, calculated using SciPy module for Python, function: *scipy.stats.linregress*. Finally, for $T_m$ determination, reproducibility and statistics are described in detail in the corresponding Methods section.

### Biosafety considerations

All procedures involving generation and handling of recombinant prions, such as PMSA, rec-PrP$^{res}$ detection, and determination of the potential infectivity of rec-PrP$^{res}$ in vitro, were carried under BSL-3 conditions given the lack of knowledge on their potential transmissibility to humans. Similarly, all experiments aimed to demonstrate

infectivity in vivo were performed in BSL-3 animal facilities. Nonetheless, as studies with some of the recombinant prions generated proceed, we have started evaluating their zoonotic potential in transgenic mouse models overexpressing human PrP^C. Those recombinant prions for which we already have completed this study with negative results (lack of transmission of clinical disease) are from then on considered, as other non-zoonotic brain-derived prions, as BSL-2 and handled accordingly.

## Reporting summary

Further information on research design is available in the Nature Portfolio Reporting Summary linked to this article.

## Data availability

The mammalian *PRNP* gene sequences obtained and used in this study have been deposited in the Genbank database (NIH genetic sequence database) under the accession codes listed in Supplementary Table 1, Supplementary Data 1, and the PrPdex (https://prpdex.com/) webpage. The uncropped scans of all the gels analyzed to get the results shown in Figs. 1 and 2, summarized also in Supplementary Data 1 and the PrPdex.com webpage, as well as the raw data used for the Misfolding Score calculations, are publicly available in Zenodo as a Dataset[44] and can be accessed through the following link: 10.5281/zenodo.10579518 (file "Fig. 1 & 2). Similarly, the raw data from inoculation studies used to generate the plots from Fig. 3, as well as uncropped blots shown in the same figure, are publicly available within the same Dataset (file "Fig. 3"). Regarding Supplementary Figs., uncropped gels shown in Supplementary Figs. 1 and 2, and the raw data used to generate the plots from Supplementary Fig. 3 are also accessible within the same Dataset from Zenodo (files "Supplementary Fig. 1", "Supplementary Fig. 2" and "Supplementary Fig. 3", respectively). Supplementary Fig. 4, that shows the main elements depicted in each individual PrPdex file (available in PrPdex and included also as Supplementary Data 1), which in turn summarize all the relevant findings from the study, share the source data with the previous figures. Except for melting temperatures calculated by Circular Dichroism, for which the raw data is available also in the same Dataset from Zenodo (file "Supplementary Fig. 4"). Finally, the source data for Supplementary Table 2, such as the number of differing amino acids with respect to bank vole PrP, the number of rec-PrP^res conformers detected, and the misfolding score, derive from the same Genbank sequences listed in Supplementary Table 1 and the gels and raw data from Figs. 1 and 2, which can be accessed in Zenodo. Additionally, the calculations performed for the predicted thermodynamic stability shown in Supplementary Table 2 are included in "Methods" section ("Computational methods, AlphaFold prediction of globular PrP structures and calculation of relative thermodynamic stability"). Additionally, the Dataset deposited in Zenodo has been included as reference[44] in the reference list.

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

## Acknowledgements

The authors would like to thank the following for their support: IKER-Basque foundation, personnel from vivarium, IT service (in particular to Sara Gómez Ramos for her assistance with the PrPdex webpage), maintenance departments of CIC bioGUNE, Neiker and IRTA-CReSA. The authors would also like to acknowledge the work from past laboratory members of the Prion Research Lab from CIC bioGUNE, that despite not directly involved in the manuscript have contributed along the years to the development of all the methods and techniques currently used in the laboratory (specially to Tomás Barrio and Leire Hervá for their efforts at the initial and end stages of the work, respectively). Finally, we would like to thank Jesús R. Requena for always useful scientific discussions and advice. The present work was partially funded by different grants awarded by "Ministerio de Economía y Competitividad" (Spanish Government), grant numbers PID2021-122201OB-C21, PID2021-1222010B-C22, PID2021-125946OB-I00 and IJC2020-045506-I, funded by MCIN/AEI /10.13039/501100011033 and co-financed by the European Regional Development Fund (ERDF), and by the Instituto de Salud Carlos III (ISCIII), grant number AC21_2/00024, to J.C. Additionally, CIC bioGUNE currently holds a Severo Ochoa Excellence accreditation, CEX2021-001136-S, also funded by MCIN/AEI /10.13039/501100011033. The funders had no role in study design, data collection and analysis, decision to publish, or preparation of the manuscript.

## Author contributions

J.C. and H.E. jointly conceived the study. Additionally, J.C. supervised the work and secured funding. C.S.T.Q., C.M.D.D., M.S.J.A., M.A.P.C., E.G.M., P.P., L.F.V., J.G.A., and E.F.M. performed all PMSA and PMCA experiments (protein production and purification, substrate preparation, PMSA reaction and PMCA reactions). C.S.T.Q. and P.P. developed and evaluated PMSA and PMCA results, with assistance of C.M.D.D. N.G.A. and G.P.N. performed genetic analysis and were in charge of introducing all sequences in GenBank. J.M.C., E.V., and M.G. carried out all bioassays, including monitoring, culling, and tissue harvesting. G.T. provided animal models and insightful feedback on the work. F.P. and G.J.O. performed all computational analyses. H.E. and J.C. wrote the manuscript with contributions from the rest of the authors.

## Competing interests

Authors H.E. and J.M.C. are employed by the commercial company ATLAS Molecular Pharma SL. This does not alter our adherence to the Journal's policies on sharing data and materials and did not influence in any way the work reported in this manuscript, given that the company had no role in study design, funding, and data analysis. The rest of the authors declare no competing interests.
