## [Peer Review File · Nature Communications]

REVIEWER COMMENTS

Reviewer #1 (Remarks to the Author):

Review of Eraña et al "A novel method for generating bona fide infectious prions de novo and its application to modelling spontaneous misfolding events in hundreds of proteins"

In this manuscript, Eraña et al introduce a methodology to amplify bona fide infectious prions de novo, based on a previously developed methodology called Protein Misfolding Shaking Amplification (PMSA). With this methodology, their goal was to survey the determinants of prion protein misfolding propensity by obtaining misfolded prion proteins from over 700 mammalian species. They then injected selected misfolded prion proteins from various species in a mouse model that is highly susceptible to prion disease replication, to validate the authenticity of the prion particles generated using their methodology. The compilation and generation of over 300 recombinant PrP proteins is a big effort. Also, the PMSA methodology has been published previously, and the data is convincing that for the strains shown (Figure 2), infectivity was ascertained.

However, there are some concerns with the manuscript:

Major concerns:

1. Supplementary Figure 1 shows 27 PrP proteins ran on an electrophoresis gel. The authors mention that uniform protein concentrations are critical factor influencing misfolding, and this is agree upon. However, this figure does not show uniform protein concentrations, at least as determined by eye, and no quantification is provided, and neither are controls. This is critical for the data presented in Figure 1, which makes statements as to how well proteins have "proneness" to misfold.

2. While this method has been published by the authors previously, because PMSA for most of the PrP sequences have not been published before, it is important to describe and show that amplification is

occurring. There is no data showing amplification steps (gels, etc) over time. Supplementary Figure 1 shows final products.

3. A big question/concern is how the "misfolding" is ascertained/defined throughout the manuscript. It is stated that "rec-PrPres" is generated by PMSA, and in Figure 1, a gel showing PK-digested rec-PrP is shown for the strains in the figure. However, it is well-known that PK-sensitive PrP^{Sc} can be infectious in some cases and can share basic structural features with PK-resistant PrP^{Sc}. Thus, lack of signal (a score of "0"), for which there is no signal in the PK-digested column, would not rule out at least infectivity. Moreover, regarding the "0" score: it is unclear how the score reflects the degree of PK-digestion efficiency, as there is no dose-dependent digestion. Also, PK digestion is not the only criteria that has been established in the field for ascertaining whether a strain is considered misfolded. Other standard assays that are common in the prion and protein misfolding fields including detergent insolubility assays, etc.

4. There is some concern in the approach followed to use only the aa in the C-terminal of PrP as it is known that intrinsically disordered regions contribute to formation of aggregates via condensate formation.

5. While the infectious prion field has been generating infectious prions for a while, it is of concern that there is no indication in the manuscript of safety practices regarding the generation of such a wide-variety and broad number of prion strains from so many different species. Chronic wasting disorder is highly contagious, not only transmissible, suggesting the possibility that other ungulate PrPs could also be contagious. And then there are all the other species generated herein. It is very concerning that there is no indication of safety procedures for the handling, and storage of these generated strains.

Minor concerns:

1. There are statements with references that do not correspond to the statements made, e.g., lines 182-185 state that "Notably, PMSA consistently and rapidly facilitated the misfolding of this rec-PrP in less than 24 h, making it our gold standard in terms of misfolding proneness, not only because its behavior in vitro, but also because its well-documented susceptibility to both sporadic and induced prion diseases in vivo (Watts et al)". But this reference does not use PMSA.

2. The methodology is not described in the main text to a degree that can be understood without having to mine the Supplemental Methods extensively. For example, how the classifications were done on Figure 1 (green versus red) is not described on the main text so it is not possible to determine what

these are based on without mining extensively through the Supplemental Methods. Moreover, for this example, it is unclear how the "misfolding proneness capacity" is depicted as binary, i.e., red or green, yet a number of "proneness" is assigned also to each protein. Thus, this needs to be explained in the text.

3. Figure 4 was never found.

Reviewer #2 (Remarks to the Author):

In the manuscript by Eraña and colleagues the authors study the spontaneous misfolding of recombinant prion proteins (rPrP) derived from virtually all mammalian species. For some of them, they demonstrate their bona fide infectivity. The amount of data is huge. These data are novel, of great interest and somehow provide information on how specific amino acids variations in PrP primary structure tune spontaneous misfolding proneness.

The experiments have been carefully performed. There is also an immense effort to make the results publicly available to the scientific community as an "encyclopaedia".

However, there are major issues that need to be discussed before appropriate for publication in Nature Communications.

1. The authors generate distinct misfolded rPrP conformers, as based on their electrophoretic signatures by western blot. This is assumed to correspond to different prion strains. The authors show that some of the generated conformers are infectious in vivo or harbour seeding activity by PMCA. However, they do not provide evidence that different strains are generated. I understand that the data are not complete, but any preliminary evidence for strain variation would reinforce the core message of the manuscript and the validity of the experimental approach.

This is especially important as the authors used for their bioassay and PMCA assay transgenic mice expressing the bank vole PrP with isoleucine at position 109. Such mice can form spontaneously prions,

although mostly when PrP is overexpressed. One has to be sure that inoculation of the different rPrP conformers is not simply accelerating a slowly developing spontaneous prionization process in these mice, with as outcome a single strain being phenotypically present.

2. Discussion section.

It is not clear to me whether specific PrP amino acids or regions influence spontaneous misfolding proneness. Are there any hot spots or regions? Could it be mapped on a figure? The main conclusion so far seems to be that certain sequences are more important than the total number of variations among mammalian PrPs.

The authors suggest that their results (from lane 540) will help predicting potential future TSE outbreaks in species. I am not sure about that according to their results.

For example, rPrP from Syrian hamster did not spontaneously misfold while this model or transgenic mice expressing hamster PrP are largely used in prion laboratories for experimental infections with prions from different species (mink, cervids, sheep, etc.). More than 10 strains have been isolated in this model. Same remark for guinea pigs or transgenic mice expressing guinea PrP which are susceptible to CJD, BSE, etc.

The opposite situation has also been found. The authors show spontaneous misfolding and infectivity of dog rPrP. However, some of the authors of the present study showed by transgenic modeling or PMCA that this species was resistant to prions. Same remark with horse and rabbit rPrP which show spontaneous conversion while little susceptibility in vivo by transgenic modelling or de novo infection.

How would the authors reconcile these seemingly discrepant data?

Reviewer #3 (Remarks to the Author):

This is a very comprehensive study of prion protein (PrP) misfolding, more specifically, the de novo misfolding of PrP to the disease-associated, proteinase K(PK)-resistant PrPres conformation. De novo formation of PrPres has always been difficult and generally not consistent. The PMSA method described in this manuscript is a breakthrough. The elimination of the sonication step, which is used in the classical PMCA assay that usually generates a high degree of variabilities, is probably the reason why PMSA is

more consistent. Using this newly established assay, authors performed a thorough analysis of 383 distinct PrPs from different mammalian species and showed that the propensity of PrP to misfold to the classic PK-resistant PrPres form appears not correlated to the overall amino acid sequence similarity, thermodynamic stability, or propensity to misfold. This is the first time that the ability of PrPs from different mammalian species to form PrPres conformers are compared. Although PMSA is still an in vitro assay and not completely ideal, this comparison is important to understand the spontaneous initiation of the pathogenic PrP forms in sporadic prion disease, at least providing a practical assay to study the process. Results from this study will also be helpful for monitoring the animal species that probably more prone to develop sporadic prion disease. The following points are raised for authors to improve their manuscript.

1. Supplementary figure 3 summarizes the major finding of this study, which is a very useful resource. But the current study just revealed de novo misfolding of various PrPs to PrPres. Another useful information, how these PrPs will perform in seeded PrP misfolding is missing. For PrPs that have spontaneously generated PK-resistant forms, seeded reactions can be easily performed. Besides homologous seeding, seeded PMSA reactions using a single vole PrPres seed could potentially reveal their propensity to misfold during cross-seeding situation, particularly with serially diluted vole PrPres seed. Both information will be very useful for predicting or preventing the potential outbreak of prion disease in animals as authors stated and would make the resource more complete.
2. The above point is highlighted by Syrian hamster PrP. Syrian hamster is an animal species widely used in prion study for a long time. Some prion strains cause disease in hamster with very shorter incubation times, suggesting a robust misfolding of Syrian hamster PrP. However, the score of Syrian hamster PrP in this study is zero. Is this PrP more prone to seeded conversion?
3. Demonstrating infectivity is important to prove that the newly generated PrPres is authentic prion and infectious. Some infectivity assays are still on-going, and it is understandable that the completion of full analyses is not practical. But the manuscript is not very clear about how many have been tested by PMCA with bank vole brain homogenate as substrate, how many have been injected into TgVole mice, and how many have developed prion disease? How many have been tested in transgenic mice expressing homologous PrP? What are the incubation time and attack rate? Although this information is included in the supplementary figure 3, it is difficult to dig them out from 380 some pages. A table would be helpful, or at least a table showing the results of the 29 PrPres tested.
4. In the current manuscript, there is no description about the solubility of these PrPs before and after PMSA?
5. Authors explained that at least one of the purposes for the purification before PMCA and inoculation into TgVole mice is to concentrate PrPres. Have authors tried undiluted samples?
6. It appears that the comparison of various PrPs to bank vole PrP was only based on the number of different amino acids. It might be important to consider the properties of amino acid, e.g. polar or non-polar, positively or negatively charged. This will obviously complicate the comparison, but it will reveal whether the change of amino acid is homologous or completely different.

7. The comparison of amino acid is only based 90-231, but full-length PrP was used for the conversion reaction. The difference in the 90-231 is probably going to affect the formation of the PrPres conformer, but it is possible that the PK-sensitive N-terminal amino acids will influence the interaction between dextran and PrP or influence the PrP conversion process. If that is the case, the N-terminal amino acid comparison would be important.

8. It is stated that the de novo formation of PrPres by PMSA is a consistent assay, but in the manuscript, there is no data or mentioning about the consistency. Is the de novo formation of PrPres by bank vole PrP 100%? How many trials have authors run? This information would establish the foundation for the systemic comparison among 383 PrPs.

9. Line 165-166, "ranging from one OR (178 amino acids) to seven OR (233 amino acids)]," Are the numbers here referred to full-length PrP? Some clarification would help.

10. Different PrPres conformers have been classified according to the size of PK-resistant fragments. Do they appear with a consistent frequency in repeat experiments? Can they be stably propagated in seeded reaction? Can they seed the conversion of bank vole PrP in PMCA?

11. A lot of results are shown in the form of individual lanes. Authors should explain how they normalize protein stains from different gels.

12. In the supplementary figure 3, there is no explanation of the "310" in PrP sequence.

13. Line 626, 642, missing references.

14. Line 33, "380 different proteins" should be "380 different prion proteins"

Point-by-point response to the reviewers' comments

Reviewer #1 (Remarks to the Author):

Review of Eraña et al “A novel method for generating bona fide infectious prions de novo and its application to modelling spontaneous misfolding events in hundreds of proteins”

In this manuscript, Eraña et al introduce a methodology to amplify bona fide infectious prions de novo, based on a previously developed methodology called Protein Misfolding Shaking Amplification (PMSA). With this methodology, their goal was to survey the determinants of prion protein misfolding propensity by obtaining misfolded prion proteins from over 700 mammalian species. They then injected selected misfolded prion proteins from various species in a mouse model that is highly susceptible to prion disease replication, to validate the authenticity of the prion particles generated using their methodology. The compilation and generation of over 300 recombinant PrP proteins is a big effort. Also, the PMSA methodology has been published previously, and the data is convincing that for the strains shown (Figure 2), infectivity was ascertained.

However, there are some concerns with the manuscript:

In the first place, the authors would like to acknowledge the effort done by the reviewer with this unusually long manuscript. We appreciate all the comments and suggestions done and we will try to address all the points raised by the reviewer.

Major concerns:

1. Supplementary Figure 1 shows 27 PrP proteins ran on an electrophoresis gel. The authors mention that uniform protein concentrations are critical factor influencing misfolding, and this is agree upon. However, this figure does not show uniform protein concentrations, at least as determined by eye, and no quantification is provided, and neither are controls. This is critical for the data presented in Figure 1, which makes statements as to how well proteins have “proneness” to misfold.

As pointed out by the reviewer, rec-PrP concentration is an important factor affecting misfolding propensity *in vitro*. And it is true that in Supplementary figure 1 some apparent differences can be observed in terms of signal intensity of the 27 distinct proteins shown. In this regard, we would like to highlight that these differences do not solely respond to slightly different concentrations, but also to the variability between the gels. With the aim of showing the original substrate preparations with which PMSA experiments were performed, we chose to show products from 27 different gels from the more than 400 done during the approximately 4 years that this project has taken. Since all these proteins were run for comparison in blocks of 6 to 12 proteins, we believe that running them all together would likely reduce the apparent differences arising from inter-gel variability. In addition, from our experience with *in vitro* prion propagation and generation techniques we know there is a range within which both, the spontaneous misfolding proneness, and propagation capacity of a certain rec-PrP does not vary significantly. Thus, taking into account the semi-quantitative nature of the technique chosen to address similar rec-PrP amounts in all substrates, we believe that the noticeable differences are still within this range in which misfolding behaviour is not significantly affected. This was already proven for recombinant prion propagation by PMSA in one of our previous publications (Eraña et al. 2018, PLoS Pathogens, doi: 10.1371/journal.ppat.1008117), which worked similarly in a range of 1 to 2 μ M of rec-PrP in the substrate, but we have not demonstrated it yet in any publication for

spontaneous misfolding. Thus, we appreciate the comment from the reviewer that gives us the opportunity to amend this.

Possibly the fact that similar amounts are needed, without requiring precise measurement is not reflected well enough in the text, where the word “uniform” was used instead of more accurate choices such as “similar”. Along this line, some modifications were introduced in the main text to clearly express the need of similar protein concentrations rather than uniform or equal. See modifications in lines 162-163:

“Before assessing the misfolding propensity of each substrate, we ensured similar protein concentrations, through electrophoresis and total protein staining.”

In addition, we have included a new supplementary figure (Supplementary figure 2) to illustrate that substrates with varying rec-PrP concentrations within a certain range (from direct to 1:5 dilution, 20% less rec-PrP concentration) exhibit similar behavior. We believe this addition will reinforce the robustness and validity of the method used throughout the study. See comment on the new supplementary figure in lines 171-173:

“Both, protein quality and concentration are relevant factors that affect misfolding proneness, although, as shown in Supplementary figure 2, within a certain range of rec-PrP concentration in the PMSA substrate (from 0.1 j.tg/j.tl to 0.025 j.tg/j.tl), results are similar.”

Also, in extended materials and methods section, a comment on this issue was introduced, in the comment box just before Tip 6, that deals with protein concentration in PMSA substrates.

Comment: The recommended storage concentration is 25 mg/ml. This selection is well-suited for subsequent substrate preparation. It's worth highlighting that protein concentration is a pivotal variable in achieving reliable results in spontaneous misfolding studies. As the rec-PrP misfolding *in vitro* is influenced by the concentration within the substrate, it is important to keep similar concentrations among the different substrates, always within a certain range as illustrated in Supplementary figure 2. While the impact of the final working concentration may vary among proteins, it exerts a more pronounced effect on those with limited misfolding capacity. Regardless, maintaining a precise record of the protein concentration at every stage of the procedure is of paramount importance.

2. While this method has been published by the authors previously, because PMSA for most of the PrP sequences have not been published before, it is important to describe and show that amplification is occurring. There is no data showing amplification steps (gels, etc) over time. Supplementary Figure 1 shows final products.

The reviewer is right on the fact that amplification steps for the new proteins used in this study are not shown in the publication. Although this data could be of interest, given the large amount of information already contained in the manuscript, we decided not to show the 4 serial PMSA rounds performed with 8 tubes each for each new protein tested, as we considered more relevant to show examples of final products. While Supplementary figure 1 shows the initial product, the substrates, the figure you are likely referring to, Figure 1, shows one of the final rec-PrP^{res} products, selected to illustrate a few of the different electrophoretic patterns encountered. Similarly, in Figure 2 we show the 4th PMSA round for 6 rec-PrPs tested. We purposely show just the final round, which is the most relevant to show the rec-PrP^{res} variability obtained for different species, which are the most relevant outcome of the experiments. However, the information regarding the rest of the PMSA rounds is coded in the upper part of each panel, following the same color code (green, rec-PrP^{res} positive and red, rec-PrP^{res} negative). While we understand that showing all 4 passages could be convenient, including gels corresponding to the total of 32 tubes handled for each species was in our opinion too much taking into account the large amount of data already contained within the manuscript, but of course, if the editors agree on the relevance of showing results from the first 3 passages, we could add a new figure with a few examples of all four passages for some of the species tested.

3. A big question/concern is how the “misfolding” is ascertained/defined throughout the manuscript. It is stated that “rec-PrPres” is generated by PMSA, and in Figure 1, a gel showing PK-digested rec-PrP is shown for the strains in the figure. However, it is well-known that PK-sensitive PrPSc can be infectious in some cases and can share basic structural features with PK-

resistant PrP^{Sc}. Thus, lack of signal (a score of “0”), for which there is no signal in the PK-digested column, would not rule out at least infectivity. Moreover, regarding the “0” score: it is unclear how the score reflects the degree of PK-digestion efficiency, as there is no dose-dependent digestion. Also, PK digestion is not the only criteria that has been established in the field for ascertaining whether a strain is considered misfolded. Other standard assays that are common in the prion and protein misfolding fields including detergent insolubility assays, etc.

We understand the concern raised by the reviewer regarding misfolding assessment criteria. And in fact, the reviewer is right on pointing out that we cannot rule out that some of the proteins considered here unable to misfold (score 0) could have misfolded into protease-sensitive forms that were not taken into account in our analysis. While the criteria for deeming a PMSA product positive in terms of rec-PrP^{res} or negative definitively needs further clarification (we failed to detail that we deemed positive all those PMSA products showing a visible (naked eye detectable) band corresponding to the equivalent digested product in brain-derived prions, that is the amyloid fiber core expanding from residues around 90 to the C-terminal end of the protein), detection of potentially PK-sensitive recombinant prions through other techniques for all the PMSA products generated resulted unfeasible. As correctly pointed out, PK resistance is not the only criteria to define misfolded prions, but it is one of the strongest indicators, and most importantly, is a simple and accurate indicator of the obtention of recombinant prions. Alternative methods, such as the detergent insolubility assay are more complex to perform with these products, since generation of amorphous aggregates showing some degree of insolubility are quite a common outcome of PMSA, specially when there is no *bona fide* misfolding, what will result likely in the need of inoculating all those samples in animal models to further clarify the result. Due to the impossibility of evaluating infectivity of all rec-PrP^{res} negative samples, we decided to limit our study to PK-resistant forms, which we agree that needs to be clearly stated in the main text. For this reason, we have introduced modifications in lines 517-526 from discussion section to clarify this issue:

“In particular, we devised a method to assess misfolding propensity of those PrP variants able to give raise to protease-resistant misfolded forms specifically, due to the screening method used based on proteinase K digestion and the criteria established. This consisted of considering rec-PrP^{res} positive, only those misfolded products in which PK-resistant fragment corresponding to the amyloid fiber core expanding from residues around 90 to the C-terminal end of the protein was detectable after electrophoresis and total protein staining with the naked eye, which in the case of the rec-PrP^{res} shown in this work is a proteolytic fragment of approximately 16 kDa. Given this approach, we cannot exclude the possibility that some of the species considered unable to misfold here, could have been misfolded into protease-sensitive forms, since these kinds of prions, although rarely, have been also describe in nature³⁶.”

4. There is some concern in the approach followed to use only the aa in the C-terminal of PrP as it is known that intrinsically disordered regions contribute to formation of aggregates via condensate formation.

Despite the amino acid differences taken into account to reduce the sequences under analysis from more than 700 mammal species to 383, were those in the C-ter region of the protein, all recombinant proteins produced and evaluated through the PMSA system contained their whole N-ter region (from residue 23 to 231 in bank vole PrP numbering). It is true that for the PrP of some species, in which differences with respect to the closest sequences were found in the N-ter, the differences in residues from 23 to 90 were neglected, and thus, for those proteins with differences limited to the first 90 amino acids only one variant was selected as representative of those species. Although, as pointed out by the reviewer, this approach involves some risk of loosing relevant amino acid variations in the N-ter of the protein, in our past experience working with recombinant prion proteins with modifications in the N-ter, we never observed dramatic changes in their behaviour with regards of *in vitro* misfolding capacity. Nonetheless, we agree that a comment on this issue is also needed to acknowledge the possibility of losing some relevant information on the most amino terminal sequence variations. See comment in discussion (lines 530-536):

“Alterations in the amino terminal extreme of the protein, despite may be important in some context for PrP misfolding³⁷ do not seemingly influence the misfolding capacity of rec-PrP in the PMSA system based on our experience with proteins with various tags or modifications in this region. However, we cannot rule out completely that for some species that were grouped together taking into consideration differences only from amino acids 90 to C-ter, there could be some difference in the spontaneous misfolding score assigned here.”

5. While the infectious prion field has been generating infectious prions for a while, it is of concern that there is no indication in the manuscript of safety practices regarding the generation of such a wide-variety and broad number of prion strains from so many different species. Chronic wasting disorder is highly contagious, not only transmissible, suggesting the possibility that other ungulate PrPs could also be contagious. And then there are all the other species generated herein. It is very concerning that there is no indication of safety procedures for the handling, and storage of these generated strains.

We agree with the reviewers’ comment on the relevance of handling these potentially hazardous new recombinant proteins under appropriate biosafety conditions and indeed, all procedures involving recombinant prions have been performed following strict biosafety guidelines and in suitable facilities. Since this is our regular way of working as a laboratory almost exclusively dedicated to prion diseases, we did not think on including such a statement. Nonetheless, we agree that this is a very relevant issue and needs to be included in the manuscript. For this reason, the following has been included at the end of Materials and Methods section (lines 853-862):

“Biosafety considerations

All procedures involving generation and handling of recombinant prions, such as PMSA, rec-PrP^{res} detection and determination of the potential infectivity of rec-PrP^{res} *in vitro*, were carried under BSL-3 conditions given the lack of knowledge on their potential transmissibility to humans. Similarly, all experiments aimed to demonstrate infectivity *in vivo* were performed in BSL-3 animal facilities. Nonetheless, as studies with some of the recombinant prions generated proceed, we have started evaluating their zoonotic potential in transgenic mouse models overexpressing human PrP^C. Those recombinant prions for which we already have completed this study with negative results (lack of transmission of clinical disease) are from then on considered, as other non-zoonotic brain-derived prions, as BSL-2 and handled accordingly.”

Minor concerns:

1. There are statements with references that do not correspond to the statements made, e.g., lines 182-185 state that “Notably, PMSA consistently and rapidly facilitated the misfolding of this rec-PrP in less than 24 h, making it our gold standard in terms of misfolding proneness, not only because its behavior *in vitro*, but also because its well-documented susceptibility to both sporadic and induced prion diseases *in vivo* (Watts et al)”. But this reference does not use PMSA.

The reviewer is right; the reference used for the statement only covers the second part of the sentence that referred to the behaviour of bank vole *in vivo*. We have included also the reference to the paper in which PMSA performed with bank vole rec-PrP is described, ref 26, as suggested.

2. The methodology is not described in the main text to a degree that can be understood without having to mine the Supplemental Methods extensively. For example, how the classifications where done on Figure 1 (green versus red) is not described on the main text so it is not possible to determine what these are based on without mining extensively through the Supplemental Methods. Moreover, for this example, it is unclear how the “misfolding proneness capacity” is depicted as binary, i.e., red or green, yet a number of “proneness” is assigned also to each protein. Thus, this needs to be explained in the text.

Indeed, due to the already long extension of the manuscript, we had to take many of the details on the procedures to Supplemental material. However, we understand that the accessibility to the information highlighted by the reviewer is important for the understanding of the method and should be included in the main text. Despite there is an explanation on the procedure in results section (lines 185-201) and we have included also a comment in this regard in response to major

concern number 3, we have added also the details on “Scoring of spontaneous misfolding proneness” section in Materials and methods of the main text (lines 684-690):

“The assessment of rec-PrP^{res} positive PMSA products (depicted as green dots or squares in figures 1, 2 and 3 and in the PrPdex files) was based on the detection after proteinase K digestion, electrophoresis and total protein staining with the naked eye of a proteolytic fragment corresponding to the equivalent digested product in brain-derived prions, that is, the amyloid fiber core expanding from residues around 90 to the C-terminal end of the protein, which in the case of most of the rec-PrP^{res} shown in this work is of around 16 kDa in size.”

3. Figure 4 was never found.

It is a mistake on our side and we apologize for it. In a previous version of the manuscript a 4th figure was included showing a real file from the PrPdex, but it was finally substituted by the example file depicted in Supplementary figure 3. The reference to the non-existing figure 4 has been removed from the text.

Reviewer #2 (Remarks to the Author):

In the manuscript by Eraña and colleagues the authors study the spontaneous misfolding of recombinant prion proteins (rPrP) derived from virtually all mammalian species. For some of them, they demonstrate their bona fide infectivity. The amount of data is huge. These data are novel, of great interest and somehow provide information on how specific amino acids variations in PrP primary structure tune spontaneous misfolding proneness.

The experiments have been carefully performed. There is also an immense effort to make the results publicly available to the scientific community as an “encyclopaedia”.

However, there are major issues that need to be discussed before appropriate for publication in Nature Communications.

We really appreciate the positive comments on our work and would like to thank the effort made by the reviewer with this unusually long manuscript. We will try to address all concerns raised by the reviewer below.

1. The authors generate distinct misfolded rPrP conformers, as based on their electrophoretic signatures by western blot. This is assumed to correspond to different prion strains. The authors show that some of the generated conformers are infectious *in vivo* or harbour seeding activity by PMCA. However, they do not provide evidence that different strains are generated. I understand that the data are not complete, but any preliminary evidence for strain variation would reinforce the core message of the manuscript and the validity of the experimental approach.

We agree with the reviewer on the fact that the potential generation of distinct strains is one of the relevant findings from our work and supports the validity of the method, since it would imply that spontaneous misfolding through PMSA is able to mimic the capacity of prion proteins to give rise spontaneously to distinct conformers. However, the large amount of novel prions and potential conformers generated in this work and the more limited availability of animals for bioassays forced us to prioritize inoculation of as many as possible recombinant prions from distinct species in order to demonstrate unequivocally that many, if not all recombinant prions generated are potentially infectious. For this reason, the analysis of potential strains (based on their distinctive electrophoretic patterns) had to be delayed and is now ongoing for several species. Apart from the practical reason exposed, this decision was taken based on the following arguments:

On the one hand, our previous publication dealing with the spontaneous misfolding of bank vole already shows that the rec-PrP^{res} generated spontaneously in PMSA showing distinctive electrophoretic patterns, resulted in distinct strains upon *in vivo* characterization (see ref. 26). Therefore, the assumption that distinct patterns may likely represent different strains for the many species analysed here, is grounded on previous evidence.

On the other hand, as we are currently working on the in-depth characterization of some of the conformers presented here from different species, we believe that some of them, at least, could

be highly suitable for further studies, for example to address specific questions about the influence of cofactors on strain characteristics. Along this line, we believe that these studies will provide enough results of interest to be published on their own, while advancing few preliminary results would reduce their value and would only be a comment in the present manuscript, which already contains a large amount of information. As an example, we have started with a more profound characterization of mouse recombinant prions and sheep (VRQ) recombinant prions and we have found notable differences in incubation periods, transmissibility of distinct host genotypes (main ovine genotypes) and histopathological features, that we hope will be published soon elsewhere. Nonetheless, and without advancing useful material for future publications, we will try to defend better our assumption of the generation of distinct strains specifically referring to the bank vole recombinant prions published previously. See new comment in lines 323-329.

“Although biochemically indistinguishable rec-PrP^{res} conformers can potentially arise into different strains upon *in vivo* inoculation, the identification of distinct electrophoretic mobility patterns serves as a clear indicator of potential structural differences that might represent different strains, as we already demonstrated for several bank vole rec-PrP^{res} conformers²⁶, also included in the final data compilation. Consequently, we systematically examined all PMSA products to identify patterns suggesting possible strain variability, which will be further analysed in the future through bioassays in order to confirm the potential strain variability obtained here.”

This is especially important as the authors used for their bioassay and PMCA assay transgenic mice expressing the bank vole PrP with isoleucine at position 109. Such mice can form spontaneously prions, although mostly when PrP is overexpressed. One has to be sure that inoculation of the different rPrP conformers is not simply accelerating a slowly developing spontaneous prionization process in these mice, with as outcome a single strain being phenotypically present.

Indeed, we are well aware of this issue with transgenic mice expressing bank vole I109I PrP. In fact, apart from the TgVole 1x line used for this study, we have experience with a mouse model expressing up to 4x the same protein, that develops said spontaneous disease invariably and can be transmitted to the 1x model easily. Therefore, although this event is a very rare occurrence in our TgVole 1x model, we have performed a very comprehensive characterization of the spontaneous disease that arises in these animals both biochemically and histopathologically. Fortunately, the spontaneous disease observed in our animals is quite easily distinguishable from other exogenous prion-induced disease in the same animals. The PrP^{Sc} that characterizes this disease is atypical, with a quite low molecular weight protease resistant fragment, very similar to that observed in atypical scrapie cases. Similarly, histopathologically, these animals present a unique affection of the hippocampus that allows its distinction from other induced prion diseases in the same model. In fact, upon the minimal suspicion of a mixed pathology (featuring any of the characteristics of the atypical disease in TgVole) we deem the inoculated rec-PrP^{res} negative as it would not fulfil our criteria for *bona fide* prion, these requiring evidences of *de novo* infection and not just acceleration of a potentially pre-existing condition. Taking all this into account, all results from bioassays are carefully analysed and for those considered infectious or even more, those we suspect that can be different strains, are only considered so when we are sure of the absence of any sign of spontaneous disease.

2. Discussion section.

It is not clear to me whether specific PrP amino acids or regions influence spontaneous misfolding proneness. Are there any hot spots or regions? Could it be mapped on a figure? The main conclusion so far seems to be that certain sequences are more important than the total number of variations among mammalian PrPs.

We are aware of the issue raised by the reviewer and agree on the fact that one of the main utilities of the present work would be the detection of specific residues, hot spots or regions that could be most influential to the spontaneous misfolding proneness of PrP. In fact, the ultimate purpose of approaching the study of as many as possible PrP variants was precisely that. However, despite

the abundance of data generated, the complexity of the question precluded obtaining any robust conclusion in this regard, apart from that correctly stated by the reviewer, that the total number of variations is irrelevant compared to the effect exerted by some regions or a few amino acids in determined sequence contexts. Even after analysing hundreds of sequences, we have found that identification of some kind of “general rules” driving spontaneous rec-PrP misfolding (or the incapacity to do so) is unfeasible for now and would be likely an overstatement. To solve this, we keep including new variants in the analysis, starting from polymorphic variants described in nature for the species studied most with regards of prion diseases (sheep, deer, human...), and following with artificial PrP sequences (single mutants, indel mutants, chimeric proteins...) specifically aimed to address if there is some critical position or domain determining spontaneous misfolding capacity regardless of neighbouring or even distant regions. Given the time that these studies will take, we decided to launch this initial approach based only on what we considered wild type PrP variants for different mammal species, as the complexity of the problem may require joint efforts from the prion research community or more sophisticated approaches such a machine-learning based analysis. For the latter option, which would be our chosen approach, more data are definitively required, which we already started to generate through the incorporation of new sequences to the assay. We have included a new comment in Discussion section to acknowledge that further analysis will be required to withdraw robust conclusions on misfolding regulator hot spots that can be generally applied to all PrP variants. See new comment in lines 558-574:

“Further conclusions, such as the determination of specific residues, hot spots or regions that could be most influential to the spontaneous misfolding proneness of PrP may be theoretically achievable based on data like the one presented here. However, due to the limitations of human visual analysis in handling large and complex datasets, at this stage, we were unable to pinpoint any position or region that could determine misfolding ability or incapability regardless of the rest of the PrP sequence or find any universal rule modulating spontaneous prion misfolding. We believe that finding general rules applicable to spontaneous PrP misfolding would require an in-depth analysis of the results through bioinformatics and machine learning approaches, what in turn, may require larger datasets and dedicated studies. For this reason, we are currently working on increasing the number of PrP variants for analysis, including polymorphic variants described in some species that are, in addition, partially related to prion disease susceptibility data *in vivo*; and *ad hoc* designed artificial PrP variants (point mutants, indel mutants, chimeric constructs) specifically aimed to address if there is some critical position or domain determining spontaneous misfolding capacity regardless of neighboring or even distant regions. Furthermore, the idea of making the method and the current dataset available to the prion research community intends to accelerate this kind of studies, given that understanding such a complex phenomenon will likely require distinct approaches and a joint effort.”

The authors suggest that their results (from lane 540) will help predicting potential future TSE outbreaks in species. I am not sure about that according to their results.

For example, rPrP from Syrian hamster did not spontaneously misfold while this model or transgenic mice expressing hamster PrP are largely used in prion laboratories for experimental infections with prions from different species (mink, cervids, sheep, etc.). More than 10 strains have been isolated in this model. Same remark for guinea pigs or transgenic mice expressing guinea PrP which are susceptible to CJD, BSE, etc.

The opposite situation has also been found. The authors show spontaneous misfolding and infectivity of dog rPrP. However, some of the authors of the present study showed by transgenic modeling or PMCA that this species was resistant to prions. Same remark with horse and rabbit rPrP which show spontaneous conversion while little susceptibility *in vivo* by transgenic modelling or *de novo* infection.

How would the authors reconcile these seemingly discrepant data?

As correctly pointed out by the reviewer, some of our results conflict with previously published data or known facts, being the example of the Syrian hamster PrP one of the most striking cases.

Indeed, it is well known that Syrian hamsters or transgenic mice expressing this protein are able to propagate several prion strains, albeit in our system it did not misfold spontaneously, receiving a score of 0. In the light of the unexpected results obtained with Syrian hamster rec-PrP, we searched for specific residues that this rec-PrP shares with other species unable to misfold and we found that all those containing Methionine at positions 112 and 139 (in Syrian hamster PrP numbering) resulted in a misfolding score of 0. Surprisingly, from more than 700 species analysed, just another 3 species share this combination of residues with Syrian hamster: *Marmota* spp, *Cynomys* spp and *Spermophilus* spp (all of them belonging to the subfamily Xerinae). Other species from the same subfamily that do not contain some of those Methionines, show a reasonable capacity to misfold. As we also knew from a previous publication that cofactors in the prion propagation media affect the resulting strain properties (Fernández-Borges N et al. Cofactors influence the biological properties of infectious recombinant prions. *Acta Neuropathol.* 2018 Feb;135(2):179-199. doi: 10.1007/s00401-017-1782-y), we hypothesized that the kind of strain that results upon misfolding in presence of dextran sulfate may not be suitable for Syrian hamster rec-PrP or any other with the two methionine residues mentioned above. To address this, we designed several Syrian hamster rec-PrP mutants that when submitted to PMSA, seem to confirm our suspicions, clearly pointing to an effect of the specific cofactor required in this study also given its role on promoting spontaneous misfolding. Our working theory is that sulfated dextran-facilitated spontaneous misfolding may preferentially lead to a determine PrP^{Sc} conformer, being this structural arrangement incompatible with those methionine residues, likely due to steric constrains in the favoured rec-PrP^{res} conformer. This cofactor-modulated misfolding is probably the reason behind the discrepancy observed also with dog rec-PrP. We previously reported its low susceptibility to the propagation of several brain-derived prion strains, as well as for rabbit. However, our major finding in these studies was that they were not completely resistant to prion infection and that their PrPs could be misfolded, although not easily. Following our cofactor-modulated misfolding theory, it seems that dog PrP can easily adopt this specific conformation, in contrast to what could happen with natural isolates tested previously as seeds to misfold dog PrP. In the case of rabbit and horse, with a misfolding score lower than 10, our results match with what was previously observed with natural isolates, implying there could be some other more strain-independent mechanism at play to explain their low misfolding susceptibility. Altogether, we are convinced that, beyond discrepancies with preexisting data, our system has unveiled the existence of PrP sequences, the misfolding of which could be highly dependent on the dominant PrP^{Sc} conformer, whether determined by a pre-formed seed or a specific cofactor. Nonetheless, we understand that this cofactor-dependence may introduce bias in our method, influencing the resulting score. In case of changing the spontaneous misfolding-assisting cofactor, the score could, at least in some cases, be different. We are currently working on establishing a similar method to evaluate misfolding proneness using other cofactors, but finding molecules different enough from dextran sulfate while exerting a similar effect in terms of promoting spontaneous misfolding *in vitro* has proven challenging. In any case, given the relevance of the issue, we introduced a new comment in Discussion section to clearly state that the cofactor used throughout the study can influence some of the results presented here. Also, we have removed the sentence stating this method could help predict future outbreaks, because given the bias introduced by a synthetic cofactor, it is an overstatement (lines 613-614). See new paragraph in Discussion section in lines 586-596:

“In the context of our method, where dextran sulfate was selected due to its misfolding-promoting effect observed for bank vole rec-PrP, it is plausible that some PrP sequences could exhibit varying misfolding propensities in the presence of other cofactors. In this regard, it is important to highlight that some of the results shown, such as the incapacity of Syrian hamster PrP to misfold, conflict with previous studies where they have been used as animal models of prion disease and demonstrated susceptibility to various prion strains⁴⁴. These discrepancies may respond to their specific behavior in presence of dextran sulfate, which may promote misfolding of PrP into a specific type of conformer, somehow different to known prion strains, and impossible to adopt for some PrP variants. This cofactor specificity, despite introducing bias in the method developed, could be however useful to further explore strain-cofactor relationship and

indicates the potential of the PMSA system for the study of strain determinants.”

Reviewer #3 (Remarks to the Author):

This is a very comprehensive study of prion protein (PrP) misfolding, more specifically, the de novo misfolding of PrP to the disease-associated, proteinase K(PK)-resistant PrPres conformation. De novo formation of PrPres has always been difficult and generally not consistent. The PMSA method described in this manuscript is a breakthrough. The elimination of the sonication step, which is used in the classical PMCA assay that usually generates a high degree of variabilities, is probably the reason why PMSA is more consistent. Using this newly established assay, authors performed a thorough analysis of 383 distinct PrPs from different mammalian species and showed that the propensity of PrP to misfold to the classic PK-resistant PrPres form appears not correlated to the overall amino acid sequence similarity, thermodynamic stability, or propensity to misfold. This is the first time that the ability of PrPs from different mammalian species to form PrPres conformers are compared. Although PMSA is still an *in vitro* assay and not completely ideal, this comparison is important to understand the spontaneous initiation of the pathogenic PrP forms in sporadic prion disease, at least providing a practical assay to study the process. Results from this study will also be helpful for monitoring the animal species that probably more prone to develop sporadic prion disease. The following points are raised for authors to improve their manuscript.

The authors would like to thank the positive comments from the reviewer and the tremendous effort done to help us improve such an unusually long manuscript.

1. Supplementary figure 3 summarizes the major finding of this study, which is a very useful resource. But the current study just revealed de novo misfolding of various PrPs to PrPres. Another useful information, how these PrPs will perform in seeded PrP misfolding is missing. For PrPs that have spontaneously generated PK-resistant forms, seeded reactions can be easily performed. Besides homologous seeding, seeded PMSA reactions using a single vole PrPres seed could potentially reveal their propensity to misfold during cross-seeding situation, particularly with serially diluted vole PrPres seed. Both information will be very useful for predicting or preventing the potential outbreak of prion disease in animals as authors stated and would make the resource more complete.

We completely agree with the reviewer regarding the analysis of seeded misfolding of all the rec-PrP variants tested for spontaneous misfolding. Although this is not included in the current manuscript, we have started an evaluation of the propagation capacity of a few of the species for which their spontaneous misfolding capabilities were determined. As an example, when initially developing this method for bank vole rec-PrP misfolding, we also assessed the propagation capacity of all recombinant seeds generated in homologous substrate by PMSA. These experiments demonstrated robust propagation in bank vole rec-PrP containing PMSA substrate by serial dilution. Although unpublished yet, we are also performing cross-transmission experiments *in vitro* with certain recombinant prions, yielding positive results. Despite being of great interest, systematic analysis of seeded misfolding capacity for each of the rec-PrP variants tested here would imply performing hundreds if not thousands of new propagation experiments and is in our opinion not as useful as it initially seems. Testing homologous seeding of all distinct rec-PrP^{res} generated in this study by serial dilution, based on our results with bank vole and a handful of other species, may not yield valuable insights into the misfolding capacity of the rec-PrP from the substrate. Our results consistently indicate that once achieved, the conformation adopted by rec-PrP^{res} is easily propagated to the homologous substrates, with observed differences being more related to distinct rec-PrP^{res} rather than the substrate itself. That is, distinct bank vole rec-PrP^{res} exhibit variations in propagation within homologous substrate, as well as the rec-PrP^{res} from other species tested. Therefore, we believe that more than being informative with regards of the misfolding capacity of the rec-PrP from the substrate, it tells something about the strain properties of the conformers generated spontaneously. While cross-seeding assays, which we believe could provide valuable insights into mimicking interspecies transmission, are theoretically intriguing, conducting them systematically is impractical due to the enormous number of possibilities in terms of seed-substrate combinations. Taking into account that only for

bank vole rec-PrP we have identified four different rec-PrP^{res} displaying differential interspecies transmissibility, the execution of thousands of experiments would be required for a systematic approach such as the one carried out for spontaneous misfolding. For this reason, we intend to include further information on the propagation capacity of select rec-PrP variants of interest in future publications addressing specific questions. Furthermore, we hope that researchers with a particular interest in certain species or barriers will be inclined to explore this avenue further.

2. The above point is highlighted by Syrian hamster PrP. Syrian hamster is an animal species widely used in prion study for a long time. Some prion strains cause disease in hamster with very shorter incubation times, suggesting a robust misfolding of Syrian hamster PrP. However, the score of Syrian hamster PrP in this study is zero. Is this PrP more prone to seeded conversion? The case of Syrian hamster rec-PrP serves as a pertinent example to illustrate the point raised in the previous comment. As correctly stated by the reviewer, despite its apparent inability to misfold in our system, the Syrian hamster is a well-established model in prion research, demonstrating susceptibility to a broad spectrum of prion strains. Nonetheless, it is well known also that it is not equally susceptible to any prion strain, being this the case for example for classical BSE and H-BSE whereas it is highly susceptible to L-BSE. In this context, the main determinants of transmission are the strain features, overriding the overall susceptibility of hamster PrP to spontaneous misfolding. If, by chance, we were to select for this kind of seeded conversion assay a rec-PrP^{res} with features similar to C-BSE or H-BSE, it would indicate that hamster PrP is not inherently more prone to seeded conversion than spontaneous conversion. In contrast if we choose seeds with transmission properties similar to L-BSE we would conclude the contrary, being the seeds the responsible for such behaviour and not hamster PrP. Thus, spontaneous misfolding capacity as ranked here would be independent from seeded propagation capacity, and this would be very difficult to assess systematically given our still limited knowledge of the recombinant seeds generated and the intricate dependence of the event on the strain characteristics of the seed. Furthermore, in relation to this issue, there is a bias introduced by our system due to the use of sulfated dextran as a cofactor, making it even more complicated to evaluate seeded propagation proneness. In the light of the unexpected results obtained with Syrian hamster rec-PrP (unable to misfold spontaneously, while being a well-known prion disease model), we searched for specific residues that this rec-PrP shares with other species unable to misfold and we found that all those containing Methionine at positions 112 and 139 (in Syrian hamster PrP numbering) resulted in a misfolding score of 0. As discussed also with reviewer 2, we noted in a previous publication that cofactors in prion propagation media influence the resulting strain properties (Fernández-Borges N et al. Cofactors influence the biological properties of infectious recombinant prions. *Acta Neuropathol.* 2018 Feb;135(2):179-199. doi: 10.1007/s00401-017-1782-y). Building on this knowledge, we hypothesized that the strains resulting from misfolding in the presence of dextran sulfate might not be suitable for Syrian hamster rec-PrP or any other with the two methionine residues mentioned above. To address this, we engineered several mutants of Syrian hamster rec-PrP. When subjected to PMSA, these mutants appear to confirm our suspicions, clearly pointing to an effect of the specific cofactor required in this study also given its role on promoting spontaneous misfolding. Our current working hypothesis is that sulfated dextran-facilitated spontaneous misfolding may preferentially lead to a determine PrP^{Sc} conformer range. This structural arrangement appears incompatible with the presence of those methionine residues, likely due to steric constraints in the favoured rec-PrP^{res} conformer. For this reason and considering that the cofactor is key to achieve high misfolding efficiency by PMSA both seeded and unseeded, we think that evaluation of seeded propagation capabilities and cross-seeding studies need to be approached little by little to respond specific questions, rather than through a systematic approach as the one presented here for spontaneous misfolding.

3. Demonstrating infectivity is important to prove that the newly generated PrP^{res} is authentic prion and infectious. Some infectivity assays are still on-going, and it is understandable that the completion of full analyses is not practical. But the manuscript is not very clear about how many have been tested by PMCA with bank vole brain homogenate as substrate, how many have been injected into TgVole mice, and how many have developed prion disease? How many have been

tested in transgenic mice expressing homologous PrP? What are the incubation time and attack rate? Although this information is included in the supplementary figure 3, it is difficult to dig them out from 380 some pages. A table would be helpful, or at least a table showing the results of the 29 PrPres tested.

The inoculation of rec-PrP^{res} generated throughout this study is a currently ongoing work, and the data included in the PrPdex files will keep growing as the bioassays progress. At the time of writing this manuscript, bioassays have been initiated for 29 distinct rec-PrP^{res}, with most of them exhibiting *bona fide* prion disease based on biochemical analysis of PrP^{Sc}. But, as pointed out, the data remain still incomplete concerning attack rates and definitive incubation periods. In light of this incompleteness, we decided not to provide a table with incomplete information at this time. We believe that such incomplete data would contribute minimally to the manuscript. The primary goal of the bioassays is to demonstrate that our system generates infectious prions, particularly in the absence of homologous models, at least in TgVole animals. Instead, we have included data from the only complete bioassays at the time of writing, those showed in Figure 3. This is to underscore the infectious nature of the PMSA products and provide an example of incubation periods and attack rates achieved in the first passage. Furthermore, new groups of animals are being inoculated every other week, and this data will be continuously incorporated to the PrPdex. Additionally, some of the most compelling results from bioassays will undoubtedly be featured in future publications specifically aimed to show findings of particular interest within some mammal families. While we recognize that a table summarizing partial information could facilitate the search of specific results, we are cautious about potential confusion arising from the inclusion of partial data from ongoing experiments. This may create the impression of incomplete attack rates or shorter incubation times than the final results would suggest. Nonetheless, to facilitate the search of this data among the 380 files of the PrPdex, we have included a new supplementary table (Supplementary Table 3) with information on the *in vitro* propagation and *in vivo* inoculation results of the 29 species for which infectivity assays are currently ongoing. See the comment included in the main text in reference to the new supplementary table 3 in lines 399-405:

“Although still ongoing for several of the rec-PrP^{res} generated, to date apart from the 4 completed assays shown in Figure 3, infectivity studies have been initiated for 29 rec-PrP^{res} from species belonging to 13 different orders either *in vitro*, *in vivo* or both (see Supplementary Table 3 containing information of currently ongoing assays). In addition, as these are completed, infectivity of other rec-PrP^{res} will be evaluated, including the results for each variant in the final data compilation in their individual misfolding files.”

4. In the current manuscript, there is no description about the solubility of these PrPs before and after PMSA?

The reviewer is right, we did not include any explicit comment on the solubility of the rec-PrP before and after PMSA. Prior to PMSA, all the rec-PrP from the substrate is soluble. In fact, as detailed in the extended materials and methods, it is critical to centrifuge the protein after dialysis for the preparation of the PMSA substrate, since insoluble amorphous aggregates interfere with the misfolding process reducing its efficiency. In contrast, after PMSA, all the misfolded product that we detect is insoluble, given that even in absence of PK digestion, there is no detectable rec-PrP signal in the PMSA product supernatant after centrifugation at 20 000 g for 15 min.

To further clarify this, specific comments have been included in the materials and methods sections in lines 642-644:

“The dialyzed protein was centrifuged at 19,000 g for 15 min at 4 °C to eliminate amorphous aggregates and the supernatant, containing only soluble rec-PrP, was used for substrate preparation.”

And lines 665-667:

“Following digestion, samples were centrifuged at 19,000 g at 4 °C for 15 min, the supernatant discarded, and the pellets, containing only PK-resistant and insoluble rec-PrP^{res}, were resuspended and washed with at least 700 µl of PBS (Fisher Bioreagents).”

5. Authors explained that at least one of the purposes for the purification before PMCA and inoculation into TgVole mice is to concentrate PrPres. Have authors tried undiluted samples? Indeed, we submitted the PMSA products to a purification by ultracentrifugation through a density gradient but only for *in vitro* infectivity assays. The inoculum for bioassays is actually the PMSA product directly diluted in PBS at 1:10. The need of purification, besides from concentrating the rec-PrP^{res} and potentially reduce the number of PMCA passages to demonstrate PrP^C misfolding by the recombinant seeds is done to avoid interference with the dextran sulfate in PMCA. During our initial studies we found inconsistent results when using directly PMSA products as seeds for PMCA with TgVole brain homogenates. After several tests, we found that eliminating or reducing the dextran from the PMSA product led to more consistent results. Therefore, to speed up the process and avoid any interference from the dextran sulfate we decided to include the purification step detailed in materials and methods section. However, in order to avoid any confusion with this issue, we included the following clarification in the corresponding Results section (lines 387-393):

“This purification step before PMCA, apart from concentrating the rec-PrP^{res} from the sample potentially reducing the number of rounds needed for misfolding of brain PrP^C, contributed to reduce the concentration of dextran sulfate from the seed, avoiding interference and artifacts observed in TgVole brain homogenate-based PMCA in the presence of high concentrations of dextran sulfate. These purified and concentrated products were used as seeds at 1:100 dilution for the first PMCA round of 24 h, up to three serial rounds were performed, second and third round seeded with a 1:10 dilution of the PMCA product from the previous round. On the contrary, for bioassay, PMSA products, simply diluted 1:10 in PBS were directly inoculated intracerebrally in TgVole animals or models expressing homologous PrP^C to the recombinant seed when available.”

6. It appears that the comparison of various PrPs to bank vole PrP was only based on the number of different amino acids. It might be important to consider the properties of amino acid, e.g. polar or non-polar, positively or negatively charged. This will obviously complicate the comparison, but it will reveal whether the change of amino acid is homologous or completely different.

We appreciate the reviewer's acknowledgment of the relevance of the differential properties of amino acids in each species compared to the bank vole PrP sequence. This information is of course of great interest to understand the potential effect of some substitutions. However, given the visual format in which the data is presented, as individual PrPdex files, conveying information about the nature of amino acids for each substitution within each file poses significant challenges. To address this, in addition to the file summarizing results obtained with each rec-PrP variant tested, the PrPdex webpage provides also the possibility to download the exact sequence of each species in FASTA format. This feature enables anyone interested in a particular sequence to conduct their own comparisons, not only with bank vole PrP but also with any other sequence of interest.

7. The comparison of amino acid is only based 90-231, but full-length PrP was used for the conversion reaction. The difference in the 90-231 is probably going to affect the formation of the PrPres conformer, but it is possible that the PK-sensitive N-terminal amino acids will influence the interaction between dextran and PrP or influence the PrP conversion process. If that is the case, the N-terminal amino acid comparison would be important.

The reviewer is right on pointing out that some of the elements in the flexible N-terminal of the PrP could have a relevant role in misfolding. However, after examined over 380 different PrP variants from more than 700 possibilities, we observed limited differences in the N-terminal. Most variations were within the OR region and had minimal impact on misfolding proneness, at least in our PMSA system. As an illustrative example, Syrian and Russian hamster exhibit a notable difference in the N-terminal, particularly in the number of OR, but they behave identically in terms of misfolding. Drawing from our past experience with recombinant prion proteins, we have observed repeatedly that while any tag in the C-terminal significantly altered misfolding capacity, the same tags in N-terminal were generally well accepted without markedly affecting misfolding. Therefore, although it is true that we may overlook some relevant positions in the N-terminal, the relatively low number of differences detected, together with their potentially negligible effects,

led us to prioritize changes in other regions. This decision, at least in this initial phase, allows us to focus on what we consider to be the potentially most relevant region.

In any case, to clarify this, we have introduced a new comment in discussion section on this regard (lines 530-536):

“Alterations in the amino terminal extreme of the protein, despite may be important in some context for PrP misfolding³⁷ do not seemingly influence the misfolding capacity of rec-PrP in the PMSA system based on our experience with proteins with various tags or modifications in this region. However, we cannot rule out completely that for some species that were grouped together taking into consideration differences only from amino acids 90 to C-ter, there could be some difference in the spontaneous misfolding score assigned here.”

8. It is stated that the de novo formation of PrPres by PMSA is a consistent assay, but in the manuscript, there is no data or mentioning about the consistency. Is the de novo formation of PrPres by bank vole PrP 100%? How many trials have authors run? This information would establish the foundation for the systemic comparison among 383 PrPs.

Indeed, the evaluation of the reproducibility of the method was conducted previously during the development of the system using bank vole rec-PrP. In our previous work (Eraña H et al. Understanding the key features of the spontaneous formation of bona fide prions through a novel methodology that enables their swift and consistent generation. *Acta Neuropathol Commun.* 2023 Sep 7;11(1):145. doi: 10.1186/s40478-023-01640-8), we showed the results of six independent experiments performed with bank vole rec-PrP, all yielding consistent outcomes with 100% of misfolding in all tested tubes. While the publication showcased six experiments, it's important to note that we have replicated similar experiments dozens of times with bank vole rec-PrP, consistently obtaining identical results in each case. Apart from repetitions with bank vole rec-PrP, we have repeated all assays initially resulting in a score of 0 for this study. In every instance, we obtained consistent results, with only a few exceptions. In some cases, upon repetition, one of the tubes at 4th PMSA round exhibited a faint positive signal, slightly modifying the spontaneous misfolding score: However, overall, we have found high reproducibility of the results. Given the relevance of the topic, we have included a new comment on reproducibility in Results section (lines 222-226):

“It is worth mentioning that despite the high reproducibility of the method was confirmed before using bank vole rec-PrP²⁶, all the rec-PrP that resulted in a misfolding score of 0 were subjected to an additional repetition of four serial PMSA rounds to confirm their incapacity for spontaneous misfolding. In the rare cases in which some tube turned out to be positive in the second repetition, the results of the latter were used to assign the final misfolding score.”

9. Line 165-166, “ranging from one OR (178 amino acids) to seven OR (233 amino acids)],” Are the numbers here referred to full-length PrP? Some clarification would help.

The numbers refer to the total number of amino acids of the full length, recombinant version of the PrP, which in bank vole numbering go from residue 23 to 231 (208 amino acids). We have introduced a comment to clarify this as suggested (lines 166-171):

“It highlights the similar concentrations, the different sizes of rec-PrP [mainly due to variations in the number of octapeptide repeats (OR), ranging from one OR (178 amino acids) to seven OR (233 amino acids). Amino acid number refers to the residues of each rec-PrP, all of them encompassing the equivalent region to that of bank vole rec-PrP, which spans from residues 23 to 231 (208 amino acids)], and the purity and correct folding, as evidenced by the absence of detectable oligomers or proteolytic fragments.”

10. Different PrPres conformers have been classified according to the size of PK-resistant fragments. Do they appear with a consistent frequency in repeat experiments? Can they be stably propagated in seeded reaction? Can they seed the conversion of bank vole PrP in PMCA?

As observed previously with bank vole rec-PrP, PMSA products with distinguishable electrophoretic mobility patterns were detected among the replicate tubes for several species. Although we dug deeper on this topic with the four conformers found for bank vole rec-PrP in our previous work (Eraña H et al. Understanding the key features of the spontaneous formation

of bona fide prions through a novel methodology that enables their swift and consistent generation. *Acta Neuropathol Commun.* 2023 Sep 7;11(1):145. doi: 10.1186/s40478-023-016408), finding that they behaved as different strains, all could be stably propagated, some were formed with a higher frequency than others throughout repeated assays and that they showed different efficiency regarding seeding capacity in PMCA (which was afterwards correlated with their incubation times *in vivo*), it would be unfeasible to perform such a comprehensive characterization of all the rec-PrP^{res} generated in this work. The study of strain characteristics for various rec-PrP^{res} generated from some species is currently ongoing in our lab and findings will be hopefully published in a near future in dedicated publications. In addition, the intent to make the method, as well as all materials resulting from this study, including distinct PMSA products, available to the prion research community aims to foster collaborative efforts in these kinds of studies.

11. A lot of results are shown in the form of individual lanes. Authors should explain how they normalize protein stains from different gels.

Given the large number of samples handled and the duration of the study of approximately four years we inevitably depended on comparing different gels to obtain conclusions. To facilitate this without the need of internal standards that would reduce the capacity of each gel, we established a rigorous protocol for sample processing, running, and staining. The same amount of PMSA product was processed always following exactly the same procedure regarding digestion, centrifugation, and posterior resuspension. Therefore, the same amount of PMSA product was loaded in the gels in each case. Electrophoresis time was also carefully controlled, as well as total protein staining that was performed systematically controlling the volume of BlueSafe used for staining and the incubation time. Furthermore, most of the gels run included undigested substrate samples for the subsequent set of species to be tested by PMSA, and since concentration of rec-PrP from the substrates was also controlled, they served also as controls for the staining reagent. Additionally, all the procedures were carried out by the same two workers to avoid inter-operator variability. However, despite normalizing initial sample amount, processing, and staining protocols, and using PMSA substrates run along the rest of the samples as controls for any dramatic failure in the process, we must assume some variability likely due to possible differences in the distinct batches of the staining reagent used. Although this variability could have some effect on the final misfolding score of some species, we believe that the strict standardization of the protocols followed and the inclusion of substrate samples in the gels were sufficient to avoid any significant issue when comparing results from different gels. In order to clarify this point, we have introduced a new comment on Materials and methods section (see lines 673-678):

“To minimize variations from comparing samples run in different gels, the same amount of PMSA product was processed in all cases and the different process times were strictly controlled, especially the staining time. Additionally, different PMSA substrates (containing similar rec-PrP concentrations) were run along the digested PMSA products in most gels, serving as internal controls of the staining reagent to avoid dramatic differences between gels.”

12. In the supplementary figure 3, there is no explanation of the “310” in PrP sequence.

As correctly pointed out by the reviewer we missed including an explanation for the 3_{10} helix structural motif that is depicted in supplementary figure 3 and in all files of the PrPdex. This motif is normally neglected within the main secondary structural elements of the PrP, but we wanted it to be represented, in case some residues of interest could emerge in those positions. We have included a comment on the secondary structural elements from the figure in the corresponding figure legend (lines 479-481):

“The central part of the file illustrates the main secondary structure elements of the PrP, the two short J3-sheets (J31 and J32), the three α -helices (α_1 , α_2 and α_3) and the two 3_{10} helices, emphasizing other potentially relevant sequences such as the J32- α_2 loop.”

13. Line 626, 642, missing references.

Apologies for the mistake, both references have been introduced.

14. Line 33, “380 different proteins” should be “380 different prion proteins”
It has been corrected as suggested by the reviewer.

REVIEWERS' COMMENTS

Reviewer #2 (Remarks to the Author):

The authors have addressed my concerns satisfactorily. Congratulations for this scientifically stimulating and huge piece of work!

Reviewer #3 (Remarks to the Author):

Authors properly addressed all the concerns raised by me and other reviewers. One minor issue is the newly added reference 37. The reference described that the N-terminal domain of PrP is required and sufficient for liquid-liquid phase separation, which is different from the misfolding of PrP described in this manuscript.

Point by point response to the reviewers' comments

REVIEWERS' COMMENTS

Reviewer #2 (Remarks to the Author):

The authors have addressed my concerns satisfactorily. Congratulations for this scientifically stimulating and huge piece of work!

The authors would like to thank again the reviewers for their thorough revision of the manuscript and the improvements suggested.

Reviewer #3 (Remarks to the Author):

Authors properly addressed all the concerns raised by me and other reviewers. One minor issue is the newly added reference 37. The reference described that the N-terminal domain of PrP is required and sufficient for liquid-liquid phase separation, which is different from the misfolding of PrP described in this manuscript.

First, we would like to thank the reviewers for their work on reviewing the manuscript, which has improved considerably from the original version. Regarding the issue with reference 37, the reviewer is right on pointing out that it may not have been the best choice on our side. When looking for references about the role of the N-ter on prion misfolding, various reports on the liquid-liquid phase separation phenomena were noticed and given their potential relevance to the field in upcoming years we considered one of them suitable to defend the potential importance of the N-ter in spontaneous misfolding. Nonetheless, we agree with the reviewer on the fact that the mechanism leading to liquid-liquid phase separation could be different from the misfolding occurring in PMSA. Therefore, to defend the potential involvement of the N-ter in prion misfolding, we decided to use the following reference: Khalife M, *et al.* Mutated but Not Deleted Ovine PrP(C) N-Terminal Polybasic Region Strongly Interferes with Prion Propagation in Transgenic Mice. *J Virol* **90**, 1638-1646 (2016), that reports the relevance of the N-ter for prion misfolding in vivo, presumably more similar to the misfolding phenomenon occurring in PMSA.